# In situ orderly self-assembly strategy affording NIR-II-J-aggregates for in vivo imaging and surgical navigation

Zhe Li[1], Ping-Zhao Liang[1], Li Xu[1], Xing-Xing Zhang[1], Ke Li[1], Qian Wu[1], Xiao-Feng Lou[1], Tian-Bing Ren [1], Lin Yuan [1] & Xiao-Bing Zhang [1] ✉

J-aggregation, an effective strategy to extend wavelength, has been considered as a promising method for constructing NIR-II fluorophores. However, due to weak intermolecular interactions, conventional J-aggregates are easily decomposed into monomers in the biological environment. Although adding external carriers could help conventional J-aggregates stabilize, such methods still suffer from high-concentration dependence and are unsuitable for activatable probes design. Besides, these carriers-assisted nanoparticles are risky of disassembly in lipophilic environment. Herein, by fusing the precipitated dye (HPQ) which has orderly self-assembly structure, onto simple hemicyanine conjugated system, we construct a series of activatable, high-stability NIR-II-J-aggregates which overcome conventional J-aggregates carrier's dependence and could in situ self-assembly in vivo. Further, we employ the NIR-II-J-aggregates probe HPQ-Zzh-B to achieve the long-term in situ imaging of tumor and precise tumor resection by NIR-II imaging navigation for reducing lung metastasis. We believe this strategy will advance the development of controllable NIR-II-J-aggregates and precise bioimaging in vivo.

Near-infrared-II (NIR-II: 1000–1700 nm) fluorescence imaging has broad application prospects due to its deep-tissue penetration, high imaging resolution, and low autofluorescence[1–14]. At this stage, the fluorophores with NIR-II emission mainly include inorganic materials (quantum dots, rare-earth materials, and nano-gold) and organic molecules. However, due to the unknown long-term toxicity, these inorganic materials are difficult to translate to the clinic[15–17]. By contrast, organic NIR-II molecules have attracted more attention for clinical research owing to their excellent biocompatibility. Up to date, organic fluorophores with NIR-II emission mainly concentrated on cyanine and benzo-bisthiadiazole core, and most of them were created with a large π-conjugate structure or strong electron-donating (D) and electron-accepting (A) groups[1,2,6,16]. Although impressive bioimaging as they have provided, these molecules with such designs inevitably suffer from some problems[6,15,17,18]. For example, NIR-II cyanines usually displayed poor chemical stability and photo-stability, and benzo-

bisthiadiazole dyes needed a tedious synthesis and were difficult to design activatable probes. Accordingly, it is of great value to explore an alternative strategy to construct the organic NIR-II fluorophore.

As an effective strategy to extend the wavelength, J-aggregation is a candidate method for constructing NIR-II organic dyes[19–27]. J-aggregation is often the slip-stacked alignment of chromophores, which leads to constructive coupling of excited-state transition dipoles. Thus, compared to their monomers, chromophore J-aggregates exhibit markedly different photophysical properties, including red-shifted absorption/emission wavelengths, increased absorbance coefficients ($\varepsilon$), and enhanced quantum yields ($Q_Y$). Also, some findings reveal that J-aggregation can improve the organic dye's chemical stability compared to their monomer and form nanoaggregates through orderly assembly, which might reduce the diffusion of fluorophores in vivo[28]. However, due to weak intermolecular interactions, the conventional J-aggregates are easily decomposed into

[1]State Key Laboratory of Chemo/Biosensing and Chemometrics, College of Chemistry and Chemical Engineering, Hunan University, Changsha 410082, China. ✉e-mail: xbzhang@hnu.edu.cn

monomers in the biological environment, leading to poor optical signal stability, which seriously hinders their bio-applications[29,30]. To solve this problem, currently, amphiphilic polymers or mesoporous silicon are utilized to encapsulate or absorb dyes[21–27], respectively. That is, nanoparticles are formed with the assistance of external carriers, which assist J-aggregates to maintain local high concentrations (Fig. 1a). Unfortunately, such methods would cause difficulties to construct activatable J-aggregates-based probes[19,20,29,30]. In addition, these nanoparticles formed in vitro sometimes may be prone to disassembly when diluted in the presence of hydrophobic biomolecules, leading to pseudosignals during bioimaging, which is harmful for obtaining accurate in situ information[31,32]. Therefore, it is still a huge challenge, but highly valuable to develop high-stability and activatable J-aggregates, in particular, the activatable NIR-II-J-aggregates with high stability.

Herein, we successfully constructed the high-stability activatable NIR-II-J-aggregates without the assistance of external carriers, which resolved traditional J-aggregates carrier's dependence satisfactorily, by utilizing rationally the H-bonding and π-π interactions of the precipitated fluorophore 2-(2′-hydroxyphenyl)-4(3H)-quinazolinone (HPQ). We reported a series of NIR-II-J-aggregates obtained by fusing an HPQ unit onto the conjugated structure of a simple hemi-cyanine dye; HPQ-Zzh was identified to be prominent (Fig. 1b). Based on the experimental results, these J-aggregates not only had good self-assembly stability and bright NIR-II emission in a water environment, but also good anti-diffusion properties similar to classical HPQs. Through hydroxyl group decoration, the fluorescence of these NIR-II-J-aggregates

could also be easily switched, indicating that they can be employed as a useful platform for probe design. More importantly, when the hydroxyl protective group on the probe is removed by the analytes, the released dye molecules are immediately self-assembled in situ to form nanoaggregates with NIR-II emission through intra- and intermolecular H-bonding and π-π interactions, which could help NIR-II-aggregates to overcome traditional J-aggregates carrier's dependence. Therefore, these NIR-II-J-aggregates could be used for developing NIR-II organic small-molecule fluorescent probes for in situ imaging. Based on these excellent properties, by applying the NIR-II-J-aggregates probe, we realized the ONOO- in situ detection in vivo and NIR-II fluorescence-mediated surgical navigation was achieved. A method for NIR-II J-aggregates which could overcome traditional J-aggregates carrier's dependence, and their corresponding probe construction was attempted in this study, as well as a candidate tool for long-term in situ NIR-II imaging-guided surgery was presented to enable precise resection in the future.

## Results

2-(2-Hydroxyphenyl)-4(3H)-quinazolinone (HPQ), a classical and well-known solid-state fluorophore (Fig. 2a), has been widely used to design fluorescent probes for detecting intracellular active species in situ[28,33–38]. Owing to their strong intra- and intermolecular H-bonding interactions, compared to normal soluble probes, HPQ-type fluorescent probes can quickly generate insoluble fluorescent products and then be deposited at the reaction site. As a result, these probes can be used for active species imaging or labeling in real-time and in situ.

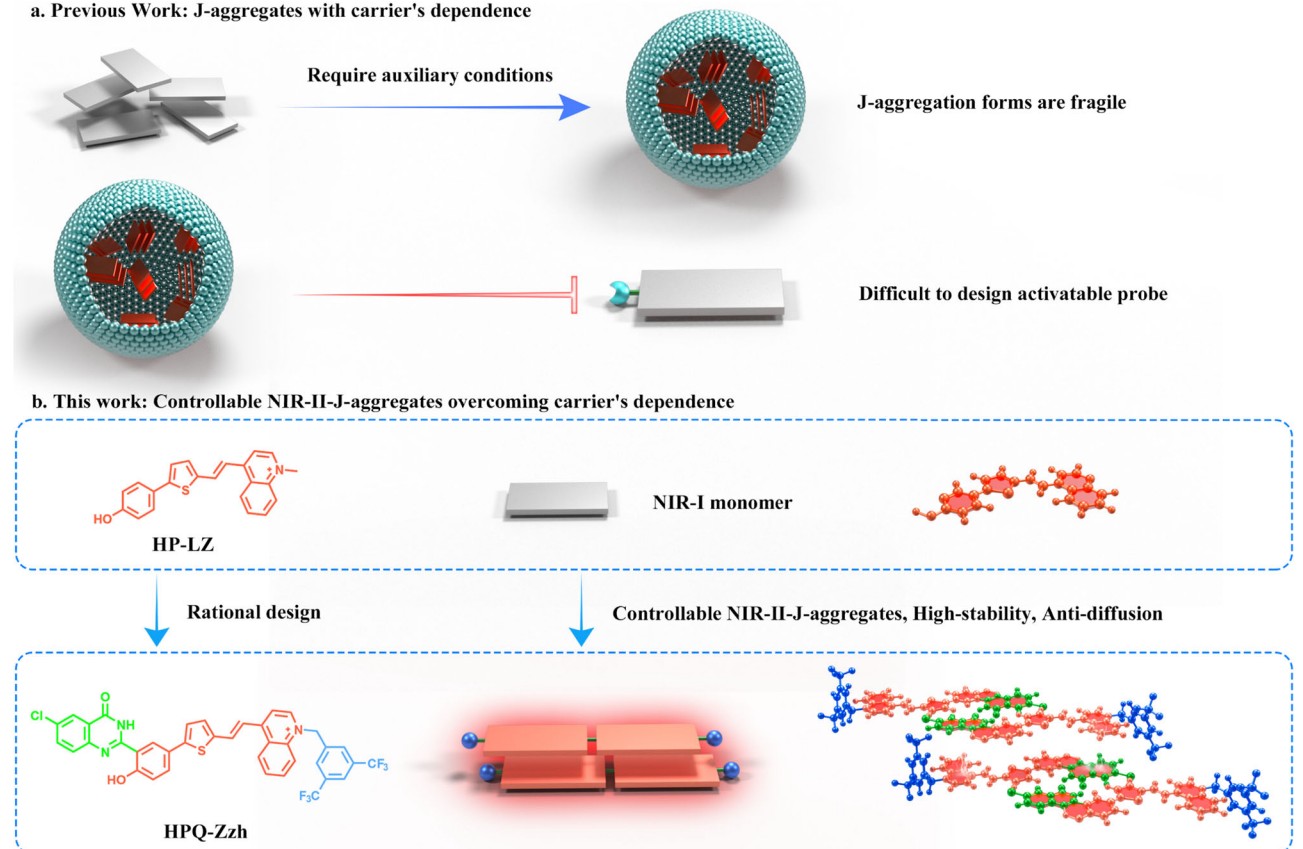

**Fig. 1 | Schematic illustration of previous work and this work. a** Traditional J-aggregation chromophore preparation method (requiring additional materials to assist stabilization). **b** An innovative method for the preparation of J-aggregates chromophores in this work (without additional materials), red part: HP-LZ, green part: HPQ unit, blue part: hydrophobic groups.

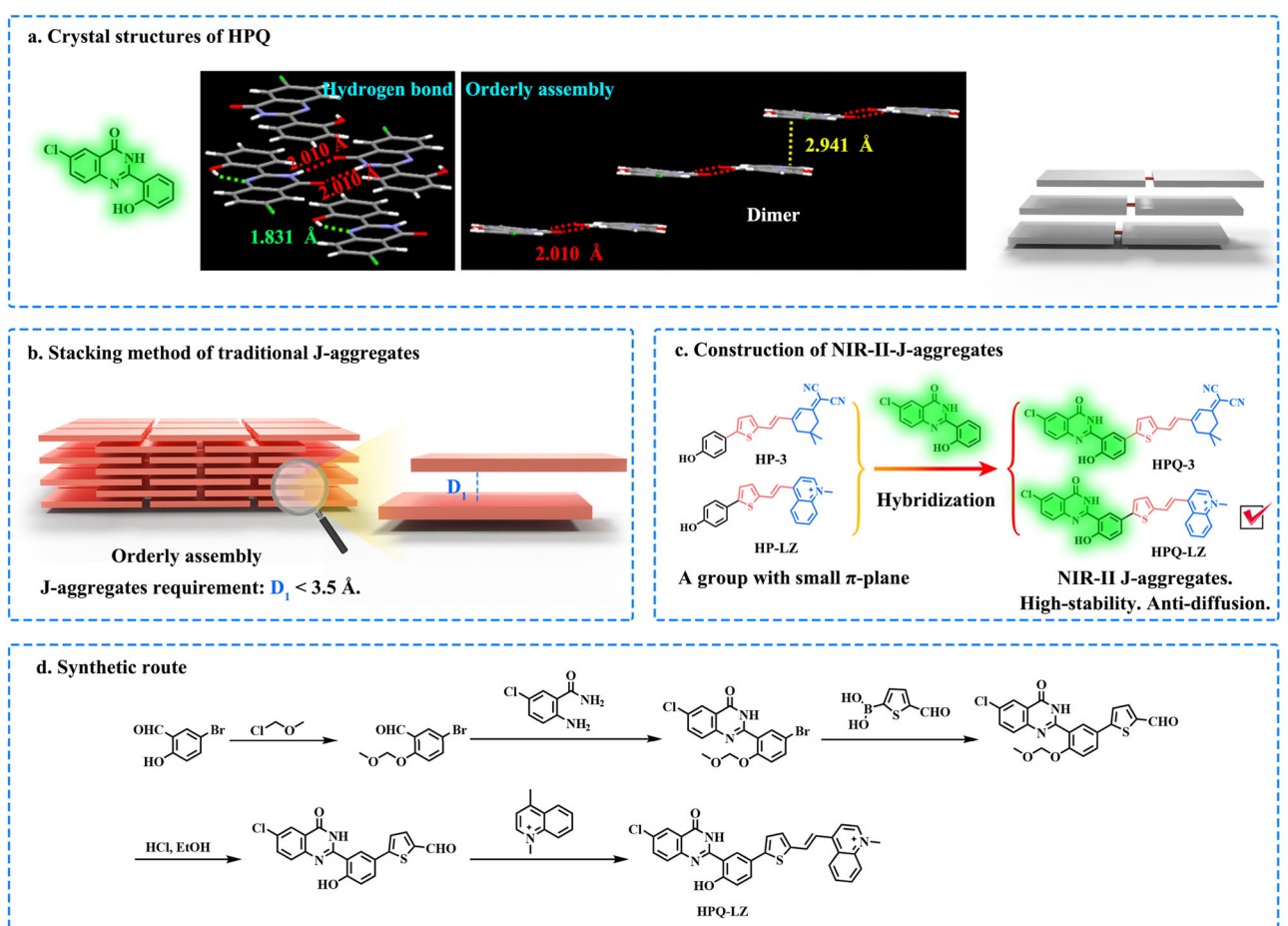

**Fig. 2 | Structure of HPQ and traditional J-aggregation strategy. a** Crystal structure of HPQ (CCDC: 2045408, the single-crystal data is from our previous article[35]) and its mechanism of in situ precipitation (strong intramolecular/ intermolecular H-bonding and π–π interactions). **b** Schematic illustration of the assembly of conventional J-aggregates. **c** The design of NIR-II-J-aggregates. **d** Synthetic route of the representative NIR-II-J-aggregates **HPQ-LZ**.

Structural and single-crystal analyses (Fig. 2a) showed that the intramolecular H-bonding interactions (OH···N: 1.831 Å) of HPQ can improve its intramolecular planarity, and the intermolecular H-bonding interactions (NH···O: 2.101 Å) can assist with dimer formation[34]. The HPQ dimer was further interconnected to form insoluble fluorescence precipitates via π–π interactions (2.941 Å). As shown in Fig. 2a, b, the π–π stacking type of HPQ dimer is very similar to that of J-aggregation (requirement: π–π distance less than 3.5 Å and orderly assembly)[39]; due to these strong and orderly intra- and intermolecular interactions including H-bonding and π-π stacking, we reason that HPQ may have the potential to assist monomer dye to form J-aggregates, which is promising to overcome the carrier's dependence.

To construct the NIR-II J-aggregates and avoid H-aggregation of the dye, the linear NIR-I emitting dyes, HP-3 (neutral molecule) and HP-LZ (positively charged molecule), were selected as the parent structures for modification (Fig. 2c). As shown in Fig. 2d, using the six-step procedure, the HPQ-mixed target dyes, HPQ-3 and HPQ-LZ, were successfully prepared. Based on the subsequent experimental results, as expected, the synthesized dye, HPQ-LZ, exhibited bright NIR-II emission in THF solution (Fig. 3a and Supplementary Fig. 2d, f). In contrast, the parent dyes, HP-LZ and HPQ-3, displayed no NIR-II emission under the same conditions (Fig. 3a, b and Supplementary Fig. 2b, c, e). Based on their absorption spectra, compared to its parent dye (HP-LZ) which had its maximal absorption at 500 nm in THF, the maximal absorption wavelength of dye, HPQ-LZ, was significantly red-shifted, reaching 750 nm (Fig. 3a and Supplementary Fig. 1c, d). This

phenomenon is consistent with the characteristics of J-aggregation and indicates that the HPQ-LZ dye may form J-aggregates in the THF solution. Of note, without the introduction of a positively charged electron-withdrawing group, the synthesized dye, HPQ-3, exhibited almost no change in the absorption wavelength compared to the original dye, HP-3 (Fig. 3b and Supplementary Fig. 1a, b). Such findings indicated that the introduction of the HPQ unit alone cannot result in the formation of J-aggregates by the linear NIR-I-emitting dye. The positively charged electron-withdrawing group is crucial for J-aggregates formation by the dye. To further verify this result, we replaced the quinoline salt with other positively charged groups, such as indole and thiazole salts, to synthesize HPQ-1 and HPQ-2, respectively. As expected, these prepared dyes (HPQ-1 and HPQ-2) had similar properties to HPQ-LZ (Supplementary Fig. 3) and exhibited obvious NIR-II emission in THF, accompanied by largely red-shifted absorptions, compared to their parent dyes, HP-1 and HP-2 (Supplementary Fig. 4). These results indicated that the positive charge and HPQ units were key elements for the formation of NIR-II-J-aggregates (Supplementary Fig. 5). To clarify the specific structure involved in the formation of NIR-II-J-aggregates, we attempted to cultivate their single crystals. Unfortunately, no single crystals were obtained, which may be due to the strong intramolecular (or intermolecular) H-bonding interactions of HPQ and the strong electrostatic interactions between the different dimers, causing tight packing of the HPQ-LZ dyes. Thus, they have poor solubility, and crystal formation in solution is difficult. In addition, according to the original HPQ's single-crystal structure (Fig. 2a), we know that HPQ is a solid-state fluorescent dye

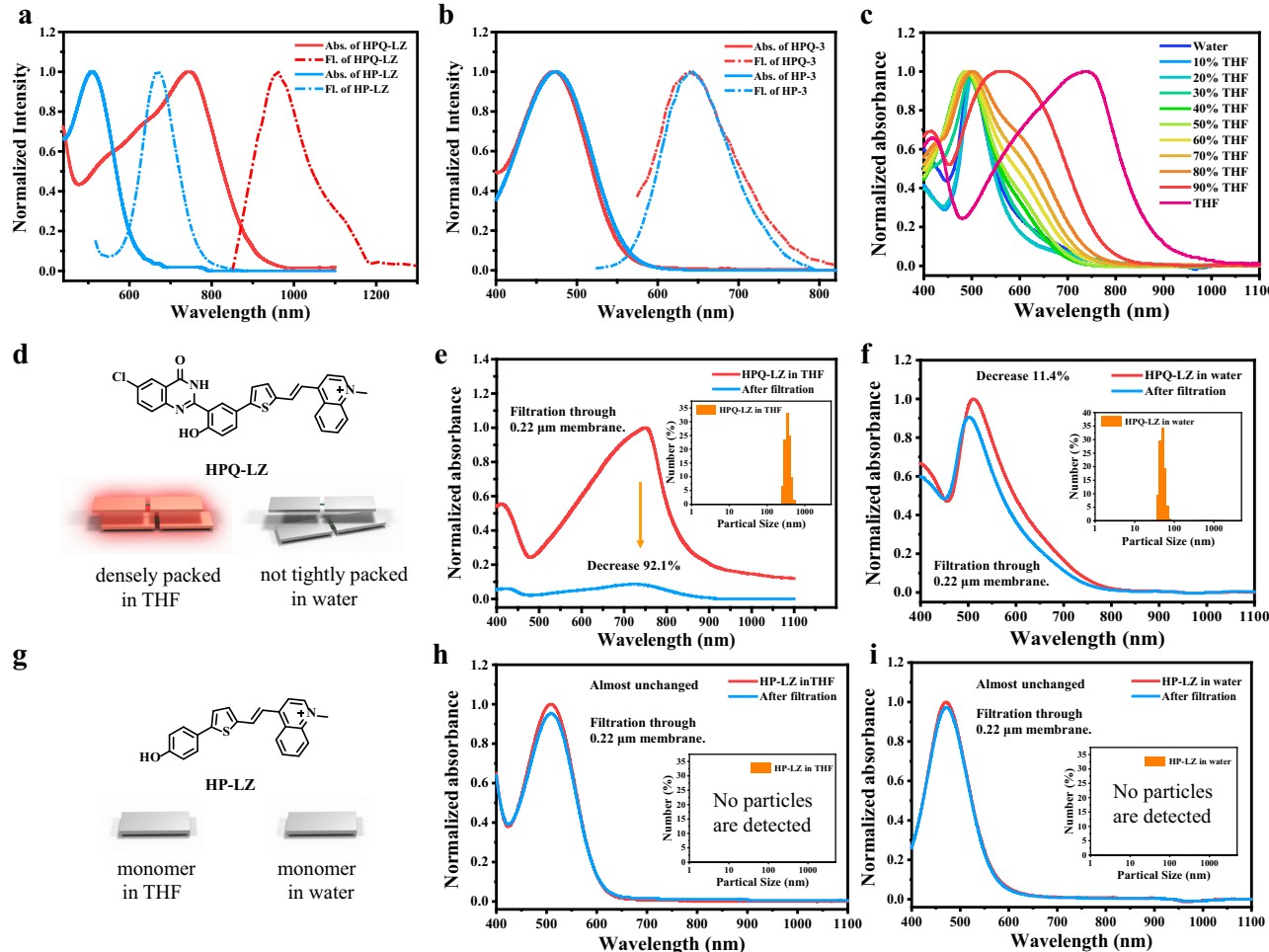

**Fig. 3 | In vitro study of spectral and particle properties. a** Normalized absorbance and fluorescence spectrum of **HPQ-LZ** and **HP-LZ** in the THF. **b** Normalized absorbance and fluorescence spectrum of **HPQ-3** and **HP-3** in the THF. **c** Normalized absorbance of **HPQ-LZ** in different ratios of THF in water. **d** The chemical structure of **HPQ-LZ**. **e** Normalized absorbance of **HPQ-LZ** in THF and absorbance after filtration through 0.22 μm membrane, the illustration showed particle size of **HPQ-LZ** in THF. **f** Normalized absorbance of **HPQ-LZ** in water and absorbance after filtration through 0.22 μm membrane, the illustration showed particle size of **HPQ-LZ** in water. **g** The chemical structure of **HP-LZ**. **h** Normalized absorbance of **HP-LZ** in THF and absorbance after filtration through 0.22 μm membrane. **i** Normalized absorbance of **HP-LZ** in water and absorbance after filtration through 0.22 μm membrane. The maximum absorbance value is 1.

based on H-bonding self-assembly and the hydrogen atom on the amide is the key to dimer formation. In order to figure out whether the NIR-II HPQs have a similar structure, we synthesized the reference compound HPQ-LZ-Me (methyl substituted for amide hydrogen) and tested its spectrum in THF. The experimental results showed that, compared with HPQ-LZ ($\lambda_{ab}/\lambda_{em}$ = 750/960 nm), HPQ-LZ-Me ($\lambda_{ab}/\lambda_{em}$ = 470/620 nm, similar to monomers) did not form NIR-II-J-aggregates (Supplementary Fig. 6), which indicated eliminating the hydrogen atom on the amide could effectively inhibit NIR-II-J-aggregates formation. On the basis, we speculated that such NIR-II-J-aggregates had a similar self-assembly mechanism to the original HPQs, thus, they were based on supramolecular assemblies of the dimers.

As the organism is a polar environment, we tested the photophysical properties of the HPQ-LZ dye in different polar solvents, especially aqueous solutions. Compared to THF, HPQ-LZ had a large decrease in NIR-II fluorescence with enhanced solvent polarity (Supplementary Fig. 2d, f). Further, the absorption wavelength of HPQ-LZ gradually blue-shifted from 750 to 500 nm (similar to the monomer's absorption) during the conversion of solvent changed from THF to water (Supplementary Fig. 1d). These results indicated that the J-aggregates of HPQ-LZ might decompose when solvent polarity increased. However, based on the DLS data, in the THF and aqueous solutions, HPQ-LZ always existed as nano-sized particles, and the

particle size in the aqueous solution (60-70 nm) was markedly smaller than that in the THF solution (>200 nm) (Fig. 3d–f). In contrast, the parent dye of HPQ-LZ, HP-LZ was soluble in both THF and aqueous solutions, and no nanoparticles were detected (Fig. 3g–i). Based on the above results, the HPQ unit can indeed assist HP-LZ to form aggregates; however, stable J-aggregates can only form in a weakly polar environment (such as THF). In polar solvents, such as water, the electrostatic interactions between the dimers of HPQ-LZ would be weakened owing to the strong interactions between the positively charged group and the polar molecules (Supplementary Fig. 7). As a result, the HPQ-LZ molecules could not pack closely, thereby forming loose and small-sized non-J-aggregation aggregates. To further verify this result, we tested the performance of HPQ-LZ in dichloromethane solution (less polar than THF). As shown in Supplementary Figs. 1, 2, when the solvent was changed from THF to DCM, HPQ-LZ still exhibited bright NIR-II emission. Such findings indicated that the non-polar environment can indeed assist HPQ-LZ with stable J-aggregates formation and emit NIR-II fluorescence. Of note, if the THF ratio was increased in the $H_2O$–THF mixtures, a clear NIR-II fluorescence turn-on, and an absorption redshift were also observed (Fig. 3c and Supplementary Figs. 8, 9), which also confirmed the above conclusion.

Based on the importance of the high brightness of the dye in the physiological environment for biological imaging, the NIR-II

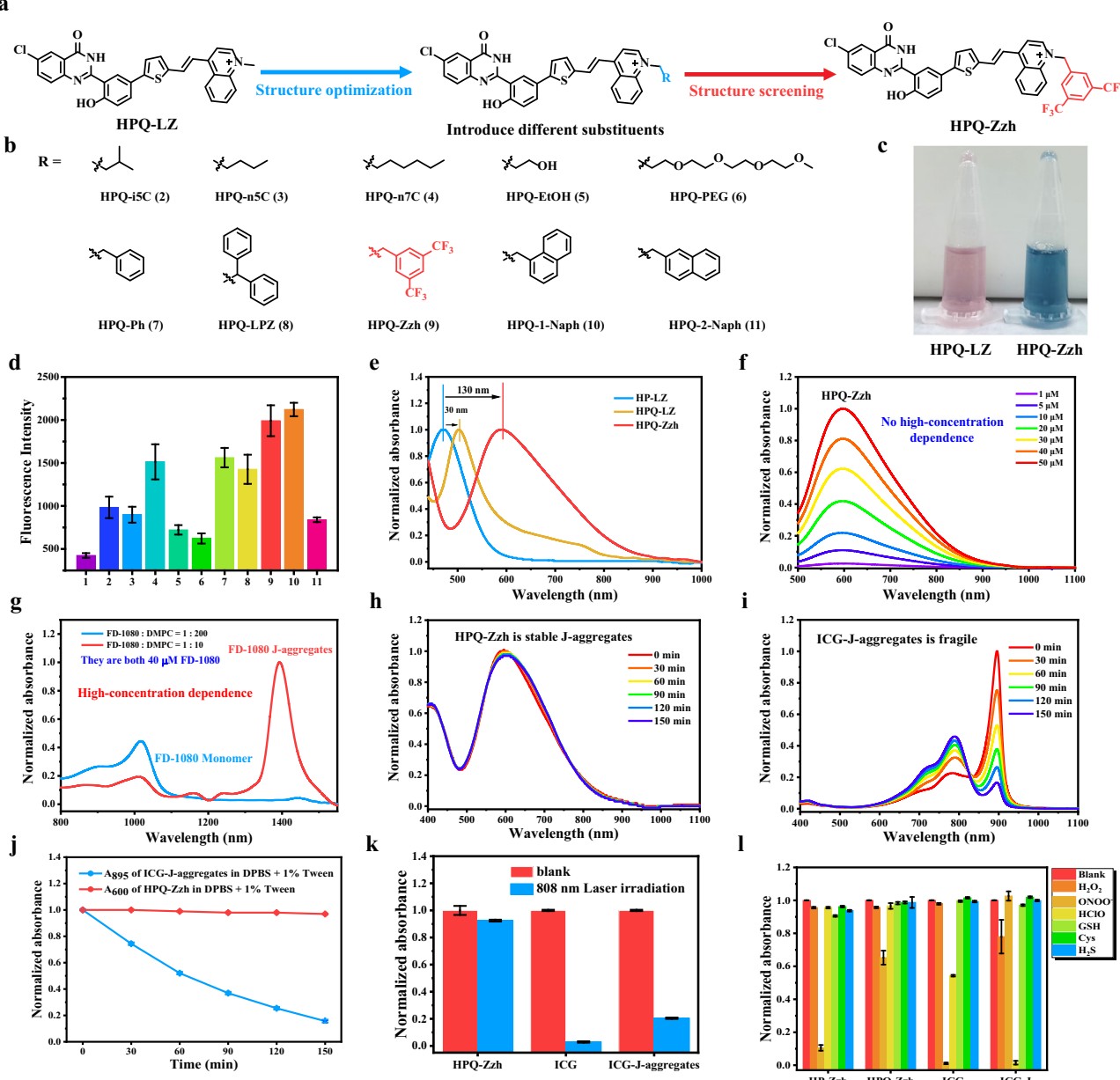

**Fig. 4 | Improved strategy of NIR-II-J-aggregates for the water environment.**
**a** Modification method. **b** Compounds with different substituents. **c** Photos of **HPQ-LZ** (left) and **HPQ-Zzh** (right) in DPBS + 1% Tween. **d** NIR-II fluorescence of all HPQ (20 μM) in DPBS + 1% Tween, (1) **HPQ-LZ**, (2) **HPQ-i5C**, (3) **HPQ-n5C**, (4) **HPQ-n7C**, (5) **HPQ-EtOH**, (6) **HPQ-PEG**, (7) **HPQ-Ph**, (8) **HPQ-LPZ**, (9) **HPQ-Zzh**, (10) **HPQ-1-Naph**, and (11) **HPQ-2-Naph**, respectively, $\lambda_{ex} = 808$ nm, Collection channel: 1000–1700 nm, Data were presented as mean ± s.d. derived from $n = 3$ independent biological samples. **e** Normalized absorbance of HP-LZ, HPQ-LZ, and HPQ-Zzh in DPBS + 1% Tween. **f** Normalized absorbance spectrum of HPQ-Zzh at different concentrations in DPBS + 10% FBS. **g** Normalized absorbance spectrum of J-aggregates with a different molar ratio of **FD-1080** to DMPC, their concentrations are both 40 μM FD-1080. **h** Normalized absorbance of **HPQ-Zzh** in DPBS + 1% Tween under different time. **i** Normalized absorbance of ICG-J-aggregates in DPBS + 1% Tween under different time. **j** Normalized absorbance of **HPQ-Zzh** and ICG-J-aggregates in DPBS + 1% Tween at different time. **k** Normalized absorbance of **HPQ-Zzh**, ICG (in DPBS + 1% Tween) and ICG-J-aggregates (in water) under 200 mW cm$^{-2}$ 808 nm laser at 0 and 30 min. Data were presented as mean ± s.d. derived from $n = 3$ independent biological samples. **l** Normalized absorbance of **HP-Zzh**, **HPQ-Zzh**, ICG (in DPBS + 1% Tween), and ICG-J-aggregates (in water) in the presence of different active substances at 60 min. $H_2O_2$ -100 μM, ONOO$^-$ -20 μM, HClO -50 μM, GSH -1 mM, Cys -50 μM, $H_2S$ -100 μM. **HP-Zzh**: $A_{520}$, **HPQ-Zzh**: $A_{600}$, ICG: $A_{800}$, ICG-J-aggregates: $A_{895}$. $A_{520}$, $A_{600}$, $A_{800}$, and $A_{895}$: Normalized absorbance of 520, 600, 800, and 895 nm. Data were presented as mean ± s.d. derived from $n = 3$ independent biological samples.

fluorescence of HPQ-LZ dyes in aqueous solutions must be enhanced. Due to the hydrophilic nature of the positively charged groups, the J-aggregation of HPQ-LZ dyes in an aqueous solution was found to be hindered. As a result, NIR-II fluorescence in aqueous solutions was very weak. To enhance the NIR-II fluorescence of the HPQ-LZ dye, the interactions between positively charged groups and water molecules must be shielded. For shielding, we introduced a series of hydrophobic groups on the positively charged groups of the HPQ-LZ dye (Fig. 4a, b

and Supplementary Fig. 10). With the introduction of the hydrophobic group, all derivatives of HPQ-LZ exhibited an obvious increase in NIR-II fluorescence in the aqueous solution compared to HPQ-LZ (Fig. 4d, Supplementary Fig. 11, and Supplementary Table 1). Particularly, HPQ-Zzh and HPQ-1-Naph displayed the best NIR-II fluorescence under the same conditions and showed an estimated five-fold stronger emission. Such results indicated that hydrophobic groups could shield the water molecules and assist HPQ-LZ to form J-aggregates, with closer packing

for NIR-II imaging in aqueous solutions. Based on their absorption spectra, HPQ-Zzh/HPQ-1-Naph had a significantly red-shifted absorption compared to HPQ-LZ in aqueous solutions, which further demonstrated their effective J-aggregations (Fig. 4e and Supplementary Fig. 12). Furthermore, it should be viewed that $\Delta\lambda_{HP-LZ, \, HPQ-Zzh}$ (130 nm) was much larger than $\Delta\lambda_{HP-LZ, \, HPQ-LZ}$ (30 nm) (Fig. 4e), which once again proved that hydrophobic groups could assist J-aggregates formation of HPQ-LZ (or its derivatives) in the physiological environment, thus emitting bright NIR-II fluorescence.

To investigate the bio-applicability of NIR-II-J-aggregates, we examined their stability under physiological conditions. Owing to its higher synthesis yield and superior brightness, HPQ-Zzh was employed to carry out follow-up experiments. As is well known, cells contain two immiscible phases, the oil phase and the water phase (including inorganic salts). DPBS is a buffered solution containing inorganic salts, which is often utilized in cell culture. While tween, a non-ionic surfactant, can form micelles in an aqueous solution, which can simulate the lipophilic structure in cells. Therefore, DPBS + 1% Tween were used to simulate the physiological environment[31]. To demonstrate that the NIR-II-J-aggregates could be utilized for bioimaging, we first investigated the self-assembly stability of HPQ-Zzh. We assessed the absorption spectra of HPQ-Zzh in DPBS + 1% Tween at different concentrations; however, the absorption peak shape of HPQ-Zzh was not found to change (Supplementary Fig. 14c, d). Notably, we added fetal bovine serum (FBS) to the experimental environment (including two situations, DPBS + 1% Tween + 10% FBS, and DPBS + 10% FBS), and the NIR-II-J-aggregates absorption peak shape remained unchanged (Fig. 4f and Supplementary Fig. 14). The experimental results showed that the absorption peaks of HPQ-Zzh at different concentrations (1–50 μM) were basically consistent, indicating that the J-aggregates of HPQ-Zzh had good self-assembly stability and lower concentration dependence (even 1 μM). As a comparison, although carriers-dependence type FD-1080 J-aggregates[23] (forming in the DMPC) displayed a stable absorption peak shape in the DPBS + 1% Tween buffer (Supplementary Fig. 15), an obvious high-concentration dependence phenomenon was observed during its preparation. As shown in Fig. 4g, the FD-1080 J-aggregates can only be yielded in a high molar ratio of FD-1080 (monomer dye) to DMPC (1:10), not formed with a low molar ratio (1:200). Furthermore, if the J-aggregates prepared without the assistance of external carriers, the FDA-approved fluorescent dye ICG affording ICG-J-aggregates quickly decomposed into ICG monomer in DPBS + 1% Tween (Fig. 4i, j), although maintaining stable in water and DPBS (Supplementary Fig. 13a). By contrast, the NIR-II-aggregates of HPQ-Zzh remained stable in all the test solutions, such as water, DPBS and DPBS + 1% Tween (Fig. 4h, j and Supplementary Fig. 13b). All these findings indicated that the J-aggregates of HPQ-Zzh had excellent stability under physiological conditions, without carrier and high-concentration dependence.

Thereafter, we tested the stability of HP-Zzh and HPQ-Zzh in DPBS + 1% Tween. Compared with the HP-Zzh monomer, HPQ-Zzh had significantly improved stability in the presence of ONOO- (Fig. 4l and Supplementary Figs. 17–19), indicating that the HPQ units would improve the stability of the NIR-II-J-aggregates. Also, HPQ-Zzh maintained good stability in the presence of $H_2O_2$, HClO, GSH, Cys, and $H_2S$ (Fig. 4l and Supplementary Fig. 19). However, ICG quickly decomposed in the presence of ONOO- and HClO (Fig. 4l and Supplementary Fig. 20) and the ICG-J-aggregates were quickly destroyed in the presence of HClO (Fig. 4l and Supplementary Fig. 21). Under 808 nm laser irradiation, the absorption peak shape of HPQ-Zzh remained unchanged, while ICG and the ICG-J-aggregates were photobleached (Fig. 4k and Supplementary Fig. 16). The findings indicated that HPQ-Zzh had better stability than ICG and the ICG-J-aggregates. All these results indicated that our NIR-II-J-aggregates assisted by the HPQ units, had good stability under physiological conditions, which was of great

significance for biological applications. In addition, we explored the internal arrangement of HPQ-Zzh through a small-angle X-ray scattering (SAXS) experiment. Experimental results showed that the SAXS atlas of HPQ-Zzh had some sharp peaks (not broad peaks, Supplementary Fig. 22a, b), which may be due to highly ordered molecular arrangement of NIR-II-J-aggregates based on H-bonding interactions (In the SAXS atlas, the more pointed peaks represent higher crystallinity and more regular molecular arrangement). Further, we characterized the morphology of the NIR-II-J-aggregates by using atomic force microscopy (AFM). The drop-casting of the EtOH solution of HPQ-Zzh J-aggregates onto the surface of a silica wafer resulted in the deposition of round cake–like aggregates (Supplementary Fig. 22c) with a length of 550 ± 20 nm, a width of 550 ± 20 nm, and a height of 85 ± 5 nm, respectively. The relatively regular surface morphology may be related to the orderly stacking structure of J-aggregates.

As the HPQ derivative dyes, HPQ-Zzh and HPQ-LZ, also exhibited bright fluorescence in their solid states, which was similar to the original HPQ dye, we proceeded to determine whether they could retain the anti-diffusion properties of HPQ. DAD-740[1,6] (log $P_{o/w}$ = 3.308, good lipophilicity) and ICG (log $P_{o/w}$ = 0.502, good hydrophilicity) were selected as the representative reference fluorophores. The diffusion of ICG and NIR-II-J-aggregates was analyzed on an agarose gel, with a diffusion of DAD-740 on filter paper as the reference experiment. Compared with ICG (slow diffusion; within 2 h) and DAD-740 (quick diffusion; within 20 min), NIR-II-J-aggregates (HPQ-LZ and HPQ-Zzh) had good anti-diffusion properties (Supplementary Fig. 23). These results indicated that the NIR-II-J-aggregates (HPQ-LZ and HPQ-Zzh) inherited the anti-diffusion properties of the original HPQ and can detect analytes in situ.

Encouraged by the above results, we then determined whether these NIR-II-J-aggregates could be used as a probe design platform by decorating their hydroxyl groups, similar to the original HPQ. Herein, HPQ-Zzh, which had a higher synthesis yield and high brightness in an aqueous solution, was selected as the signal reporter, while phenylboronic acid pinacol ester was selected as the recognition group to develop the representative probe, HPQ-Zzh-B, for ONOO- detection (Fig. 5a). As shown in Fig. 5c, d, HPQ-Zzh-B was essentially non-fluorescent (NIR-II fluorescence) owing to the protection of hydroxyl group. However, with the addition of ONOO-, NIR-II fluorescence was significantly enhanced (Fig. 5d). These findings indicated that similar to the original HPQ, the intra- and intermolecular H-bonding interactions of HPQ-Zzh can be disrupted by the hydroxyl group. No J-aggregates were formed with HPQ-Zzh-B, and no NIR-II fluorescence was emitted. During the reaction with ONOO-, the hydroxyl group of the HPQ-Zzh-B probe was released, resulting in the J-aggregation assembly of the HPQ-Zzh dye and the release of NIR-II fluorescence. For further verification, the absorption spectra of the HPQ-Zzh-B probe was tested. As shown in Fig. 5b, the HPQ-Zzh-B probe exhibited a short absorption at 500 nm. However, with the addition of ONOO-, the absorption of the probe solution was significantly red-shifted to 605 nm, which agreed well with the absorption of the J-aggregates of HPQ-Zzh (Fig. 4e). In addition, to confirm the ONOO- responsiveness of HPQ-Zzh-B, we tested the absorbance of HPQ-LZ and HPQ-Zzh in the presence (and absence) of ONOO- as the control experiments (Supplementary Fig. 24). The results showed that, different from HPQ-Zzh-B, ONOO- did not cause the absorption redshift of HPQ-LZ and HPQ-Zzh, which further indicated that the response between HPQ-Zzh-B and ONOO- was specific. Based on these results, regulating the hydroxyl group could control the J-aggregation of HPQ-Zzh and further switch its NIR-II fluorescence OFF-ON. Thus, similar to the classic HPQ, these NIR-II-emitting J-aggregates (HPQ-Zzh) could be used as platforms to design NIR-II fluorescent probes by modifying the hydroxyl group.

To verify the biological application potential of HPQ-Zzh-B, selectivity experiments were conducted. As shown in Fig. 5e and

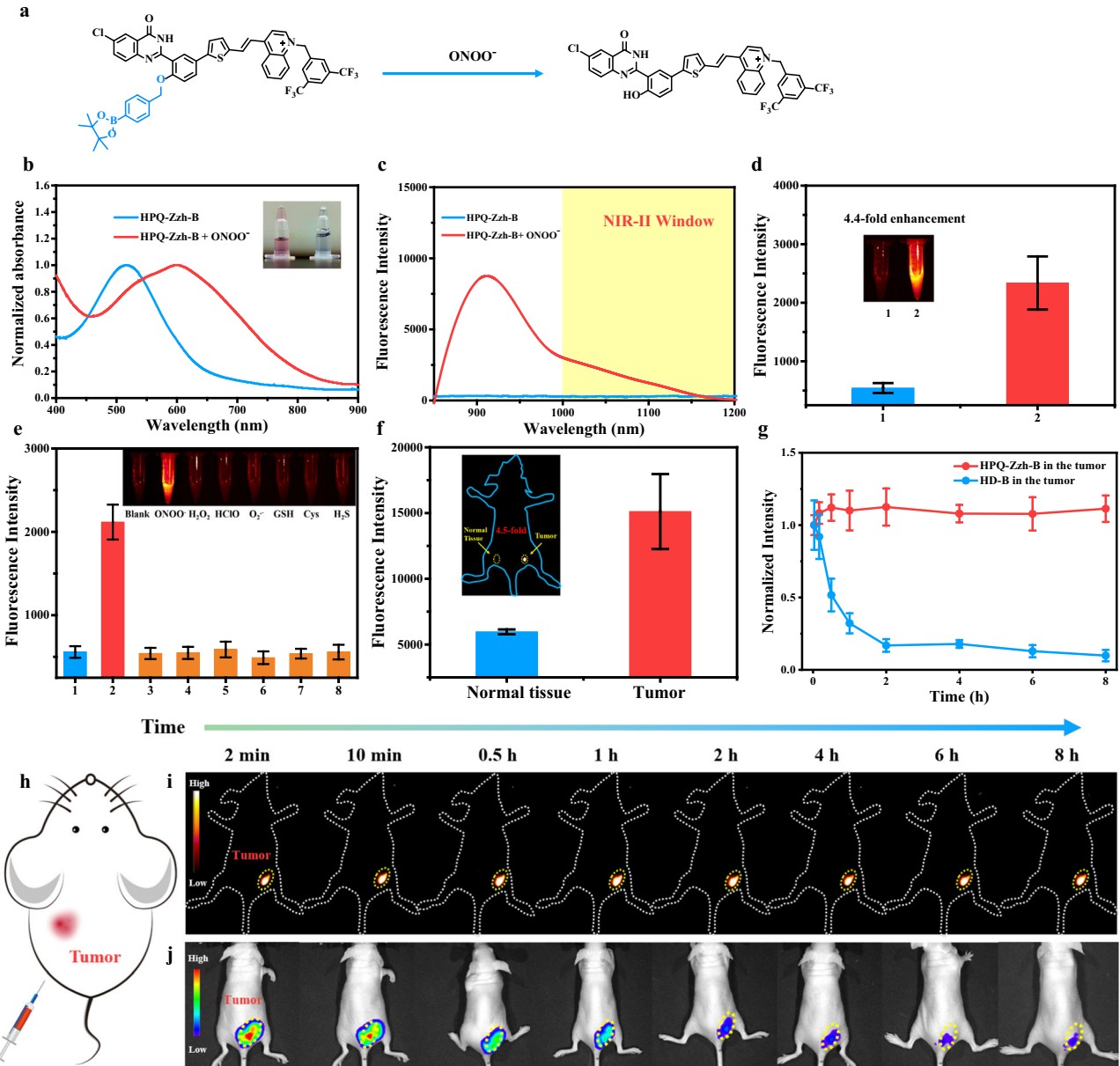

**Fig. 5 | Spectroscopy experiments and in vivo anti-diffusion experiments of HPQ-Zzh-B. a** Schematic diagram that **HPQ-Zzh-B** reacted with ONOO⁻. **b** Normalized absorbance of **HPQ-Zzh-B** and **HPQ-Zzh-B** with ONOO⁻ in DPBS + 1% Tween. **c** Fluorescence spectrum of 10 µM **HPQ-Zzh-B** and 10 µM **HPQ-Zzh-B** with 5 µM ONOO⁻ in DPBS + 1% Tween. **d** NIR-II fluorescence of 10 µM **HPQ-Zzh-B** (1) and 10 µM **HPQ-Zzh-B** with 5 µM ONOO⁻ (2) in DPBS + 1% Tween, respectively, the illustration showed their NIR-II fluorescence, left: **HPQ-Zzh-B**, right: **HPQ-LZ-B** + ONOO⁻. Data were presented as mean ± s.d. derived from $n = 3$ independent biological samples. **e** NIR-II fluorescence of **HPQ-Zzh-B** (10 µM) to various analytes (1. blank, 2. 5 µM ONOO⁻, 3. 100 µM $H_2O_2$, 4. 50 µM HOCl, 5. 20 µM $O_2^{-}$, 6. 3 mM GSH, 7. 100 µM Cys, 8. 100 µM $H_2S$.) $\lambda_{ex} = 808$ nm, collection channel: 1000–1700 nm. Data were presented as mean ± s.d. derived from $n = 3$ independent

biological samples. **f** NIR-II fluorescence of **HPQ-Zzh-B** (Left, 1) and **HPQ-Zzh-B** inside the tumor (Right, 2) in vivo, respectively, $\lambda_{ex} = 808$ nm, collection channel: 1000–1700 nm. The illustration is NIR-II imaging schematic diagram for **f**, **HPQ-Zzh-B**: 100 µM, 25 µL. Data were presented as mean ± s.d. derived from $n = 3$ independent biological samples. **g** Line diagram for **i** and **j**. Data were presented as mean ± s.d. derived from $n = 3$ independent biological samples. **h** Anti-diffusion experiments in the tumor area, schematic diagram of anti-diffusion experiment in vivo. **i** **HPQ-Zzh-B**, $\lambda_{ex} = 808$ nm, collection channel: 1000–1700 nm. Anti-diffusion experiments in the tumor area based on NIR-II fluorescence. **j** **HD-B**, $\lambda_{ex} = 640$ nm, collection channel: 690–770 nm, 100 µM, 25 µL. Anti-diffusion experiments in the tumor area based on NIR-I fluorescence.

Supplementary Fig. 25, HPQ-Zzh-B exhibited high selectivity to ONOO⁻ relative to other biological species, such as $H_2O_2$, HOCl, $O_2^{-}$, GSH, Cys, and $H_2S$. We opted to apply the probe to the image in the living body. After HPQ-Zzh-B reacted with ONOO⁻ in vivo, its NIR-II fluorescence displayed a 2.1-fold enhancement (Supplementary Fig. 26). These results indicated that HPQ-Zzh-B could be used as an efficient NIR-II bioimaging probe for ONOO⁻ detection.

As the level of reactive oxygen species (ROS) in tumors is markedly higher than that in normal tissues[40], we investigated whether the

anti-diffusion NIR-II probe, HPQ-Zzh-B, could be used for tumor imaging, especially long-term in situ mouse tumor imaging. As shown in Fig. 5f, after the administration of HPQ-Zzh-B, NIR-II fluorescence in mouse tumors increased significantly within 10 min. However, normal tissue with the injected probe displayed almost no fluorescence. Such findings indicated that HPQ-Zzh-B can be used to identify tumor tissues in vivo by ONOO⁻ detection. Of note, following the response to ONOO⁻, the fluorescence intensity of the HPQ-Zzh-B probe remained unchanged within 8 h (Fig. 5g, i). HD-B probe (HD dye modified with

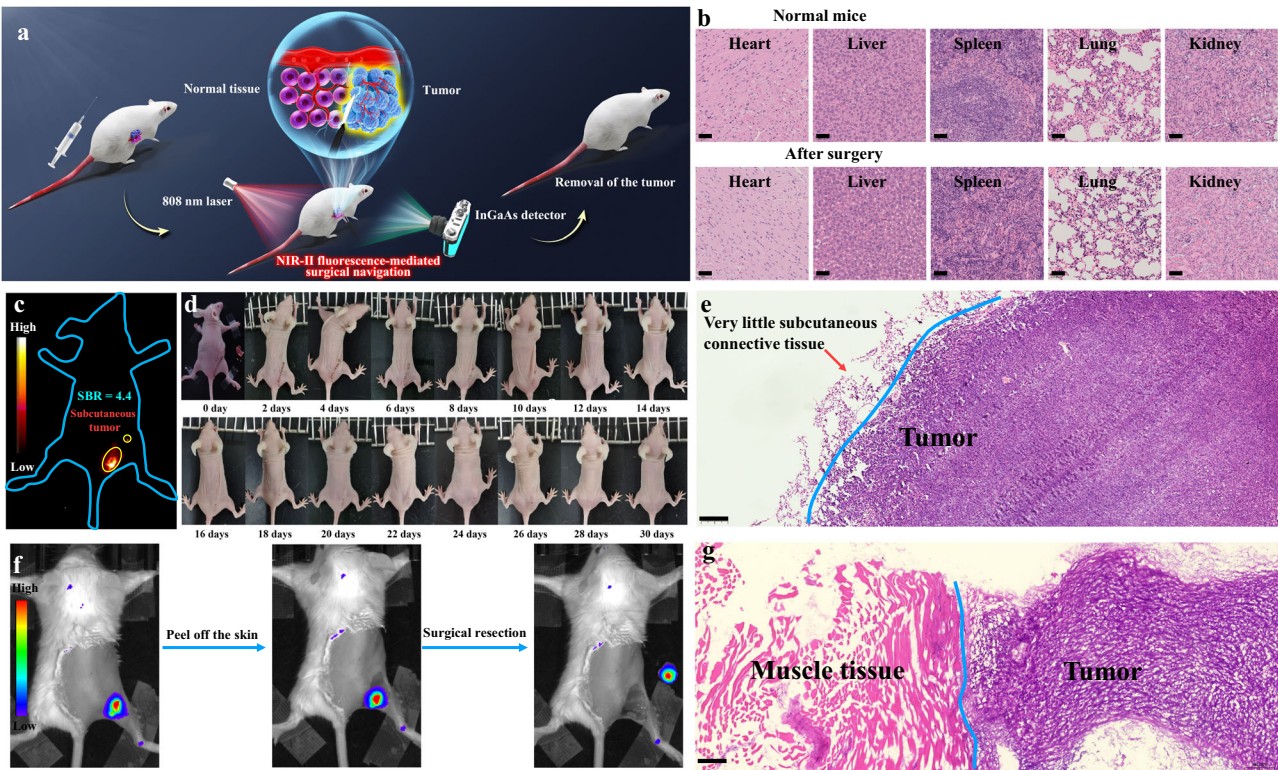

**Fig. 6 | Surgical navigation for subcutaneous tumors and recovery assessments. a** Schematic diagram of NIR-II fluorescence-mediated surgical navigation. **b** Histology studies of normal mice and tumor-removal mice for different organs, scale bar: 50 µm. The histological studies indicated that the surgical resection did not damage any organs of the mice. The experiment was repeated three times independently, with similar results. **c** NIR-II imaging in vivo before surgical navigation with **HPQ-Zzh-B**, $\lambda_{ex}$ = 808 nm, collection channel: 1000–1700 nm. **d** Photos of mice at different time (every 2 days) after surgery. The experimental mice gradually recovered health. **e** Histology studies of tumor resected after NIR-II fluorescence-mediated surgery for a subcutaneous tumor, scale bar: 200 µm. The experiment was repeated three times independently, with similar results. After

surgical navigation, the subcutaneous tumor had only a small amount of subcutaneous connective tissue. The histological studies indicated that NIR-II fluorescence, based on **HPQ-Zzh-B**, could effectively distinguish the boundary between normal and tumor tissues. **f** NIR-I imaging in vivo before and after surgical navigation with **HD-B**, $\lambda_{ex}$ = 640 nm, collection channel: 690–770 nm. **g** Histology studies of tumor resected after NIR-I fluorescence-mediated surgery for subcutaneous tumor, scale bar: 200 µm. The experiment was repeated three times independently, with similar results. After surgical navigation, the resected tissue contained not only a large amount of tumor tissue, but also a large amount of muscle tissue. The histological studies indicated that NIR-I fluorescence, based on **HD-B**, could not effectively distinguish the boundary between normal and tumor tissues.

Borate group) is a kind of widely used NIR-I ROS probe (Supplementary Fig. 27)[41]. Different from HPQ-Zzh-B, under the same conditions, HD-B rapidly spread in mice, and the fluorescence in the tumor rapidly decreased (Fig. 5g, j). These results indicated that HPQ-Zzh-B had excellent anti-diffusion performance in the living body and could be used for long-term in situ imaging in vivo in tumor.

Cancer is one of the main diseases that threaten human health. Currently, the treatment of solid tumors is primarily based on surgical resection[42–48]. The main purpose of tumor resection surgery is to completely remove the primary lesions and metastatic tumors, while minimizing damage to normal tissues. However, owing to the lack of in situ precipitation ability and easy diffusion to adjacent tissues, it is difficult to accurately identify the boundaries of tumor tissues using traditional fluorescent probes. As a result, surgical navigation using these probes often leads to excessive excision and damage to the adjacent vital organs or normal tissues.

Owing to the excellent anti-diffusion performance of HPQ-Zzh and the remarkable responses of its probe (HPQ-Zzh-B) in tumors, we applied HPQ-Zzh-B to fluorescence image-guided cancer surgery. According to the clinical strategy of surgical resection of breast cancer, we injected the probe, HPQ-Zzh-B, specifically into the tumor to delimit the boundary (4T1 tumor-bearing mice, subcutaneous tumor; Supplementary Fig. 28). As HPQ-Zzh-B can maintain a stable signal for 1–8 h in tumors (Fig. 5i and Supplementary Fig. 29), 2 h after probe injection was selected as the surgical resection window. As shown in

Fig. 6c, after the administration of HPQ-Zzh-B, the tumor could precisely light up in situ. Further, its signal-to-background ratio (SBR) reached up to 4.4 for 8 h (Supplementary Fig. 29). Subsequently, the obvious NIR-II fluorescent tumor tissue of the mouse was resected with a scalpel. Notably, owing to the reduced background fluorescence of NIR-II imaging and the excellent anti-diffusion properties of the probe, the boundary of the tumor was always clearly delimited throughout the operation (Supplementary Movie 1). After surgical navigation, the subcutaneous tumor contained only a small amount of subcutaneous connective tissue (Fig. 6e and Supplementary Fig. 30). Histological studies indicated that NIR-II fluorescence effectively distinguished the boundary between the normal and tumor tissues. Thus, all tumors could be accurately removed. Based on postoperative recovery, no mice had recurrent tumors (Fig. 6d), which further demonstrated accurate resection. Histological studies also revealed that surgical resection did not damage any organs in mice, which had fully recovered after surgical navigation (Fig. 6b). For comparison, we performed the same surgical navigation experiment with the HD-B probe (i.e., the obvious NIR-I fluorescent tumor tissue of the mouse was resected with a scalpel, as shown in Fig. 6f). In contrast to HPQ-Zzh-B, owing to the low resolution of NIR-I imaging and easy diffusion of HD-B, the resected tissue contained not only a large amount of tumor tissue, but also a large amount of muscle tissue after the surgical navigation (Fig. 6g and Supplementary Fig. 31), indicating that NIR-I fluorescence based on HD-B could not distinguish between normal and tumor

tissues. These results confirmed that the HPQ-Zzh-B probe derived from the NIR-II-J-aggregates fluorochrome can be used as an efficient tool for long-term in situ imaging-guided surgery.

Although the imaging and surgical resection of subcutaneous tumors can reproduce the surgical navigation process, they have certain limitations in guiding clinical surgery. To align with clinical practice and compare the integrity of surgical resection of tumors under the navigation and non-probe tracking, we performed primary tumor implantation and surgical resection. The need for surgically guided resection with probe imaging was assessed by comparing the completeness of tumor resection and potential lung metastases. Based on the excellent performance of HPQ-Zzh-B during surgical navigation for the removal of subcutaneous tumors, HPQ-Zzh-B was used to remove the primary tumor by NIR-II fluorescence-mediated surgical navigation. We injected HPQ- Zzh-B specifically into the tumor to delimit the boundary (4T1 tumor-bearing mice; primary tumor, Fig. 7b). Due to the good imaging properties of NIR-II fluorescence, surgical removal was successfully performed (Supplementary Movie 2). Histological studies revealed that after surgical navigation, the primary tumor had only a small amount of fat pad gland (Fig. 7c and Supplementary Fig. 33). Such findings indicated that NIR-II fluorescence effectively distinguished the boundary between the normal and tumor tissues. Subsequently, mice after surgery and the same batch of primary-tumor mice were raised for another three weeks (Fig. 7a). Thereafter, the experimental mice were sacrificed. The spleen volume of surgically resected mice was found to be almost the same as that of normal mice, and markedly smaller than that of mice with primary tumors (Fig. 7e). Histological studies indicated that the spleens of mice with primary tumors had a vigorous immune response while those of mice resected by NIR-II fluorescence-mediated surgery were similar to those of normal mice (Fig. 7f). The results of India ink staining lung experiments also showed that pulmonary tumorous nodes appeared in the lungs of tumor mice, but were not found in normal mice and mice subjected to surgical resection (Fig. 7g), which was similar to the histological results (Fig. 7j). These results indicated that NIR-II surgical navigation based on HPQ-Zzh-B effectively reduced tumor metastasis. As well known, CD206 is a protein associated with tumor metastasis, and an increase in its content indicates tumor metastasis[49]. The CD206 immunohistochemical experiments showed that the lungs of mice with primary tumor had obvious metastasis (Fig. 7i, k), whereas those of normal mice and mice subjected to surgical resection had no metastasis. According to the TUNEL assay, lungs from surgically resected mice had a lower proportion of apoptotic cells at the same time point (Fig. 7d, h) than those from mice with primary tumor. The above experimental results indicated that surgical navigation of the primary tumor based on HPQ-Zzh-B could effectively remove the tumor, markedly reducing the metastasis of tumor cells to the lung. Further, no residual tiny tumor foci were presented to cause the metastasis of 4T1 cells. These results indicated that our NIR-II-J-aggregates had good potential for clinical applications.

## Discussion

Fluorescence probes for in vivo tumor imaging usually have limited spatial resolution, tissue penetration depth, and easy-diffusion properties. Accordingly, these probes, which emit in the first NIR window (NIR-I: 650–900 nm), cannot fully meet the requirements of clinical translation. In contrast to the easy-diffusion NIR-I probes, NIR-II fluorophores with anti-diffusion properties are more suitable for in vivo tumor imaging. In 2021, our group reported an HPQ analog (HYPQ), which was used to design an anti-diffusion probe HYPQG for in situ imaging of the membrane-associated enzyme, glutamyl

transpeptidase (GGT). However, the absorption and emission wavelengths of HYPQ are less than 600 and 700 nm, respectively, which limits the penetration of biological tissues and the signal-to-background ratios (SBRs) for in vivo imaging. In contrast, NIR-II (1000–1700 nm) imaging has attracted widespread interest in the past decade owing to its deeper penetration of biological tissues, reduced autofluorescence, and improved SBRs. In this study, we introduced a positively charged electron-withdrawing group to transform existing HPQ into proper J-aggregates-based anti-diffusion NIR-II HPQs scaffolds (HPQ-LZ and its derivatives, $\lambda_{ex}/\lambda_{em} = 808/960$ nm) for NIR-II imaging. Through the regulation of the hydroxyl group, an anti-diffusion activatable NIR-II probe HPQ-Zzh-B was developed. Using the probe HPQ-Zzh-B, we not only long-term visualized the tumor, but also realized in situ NIR-II imaging-guided surgery.

In summary, by fusing an HPQ unit with strong intra- and intermolecular H-bonding interactions on the conjugated structure of the dye, an innovative strategy was established to develop stable and controllable NIR-II-J-aggregates which could overcome the conventional J-aggregates carrier's dependence. Compared to the traditional J-aggregates formed by polymer or hollow mesoporous silica encapsulation, the NIR-II-J-aggregates can not only exist stably in aqueous solutions without the aid of foreign carriers, but also exhibit controllable assembly and optical performance. Similar to the original HPQ dyes, these J-aggregates with HPQ units displayed anti-diffusion properties and bright solid-state fluorescence. Owing to such features, these NIR-II fluorophores are particularly attractive for long-term in situ imaging in biological research. To demonstrate their application, we further synthesized the activatable NIR-II $ONOO^-$ probe, HPQ-Zzh-B, based on these J-aggregates. In principle, the probe (hydroxyl-protected HPQ-cyanine) dissolved well under biological conditions and exhibited almost no fluorescence. However, when the hydroxyl protective group on the probe was removed by the analytes, the released dye molecules immediately formed a large planar dimer through intra- and intermolecular H-bonding interactions, which was further orderly self-assembled to afford J-aggregates with strong NIR-II fluorescence through π-π stacking. Using this probe, long-term in situ imaging of the tumor in vivo was not only achieved, but also precise tumor resection with NIR-II imaging navigation, which could effectively reduce lung metastasis. Our strategy would provide a means to construct a wide variety of high-stability and anti-diffusion NIR-II-J-aggregates conquering the conventional J-aggregates carrier's dependence and stimulate interest in the future design of activatable NIR-II probes for long-term in situ imaging in vivo.

## Methods

### In vivo NIR-II imaging studies

All NIR-II images were collected on a self-built small animal imaging system with 640 × 512 pixel 2D InGaAs NIRvana CCD camera (The laser light source was provided by Hong Kong Electronics Company, Light source model: FC-808-30, Excitation wavelength: 808 nm, Energy density: 200 mW cm$^{-2}$, Collection wavelength: 1000–1700 nm). Images were processed with the LightField imaging software and Image J (1.4.3.67).

### Data analysis

All statistical graphs, absorption spectrum, and fluorescent spectra were analyzed with OriginLab 2019. NMR files were analyzed with MestReNova. Mass spectrum files were analyzed with flexAnalysis.

### Ethical statement

All animal experiments were conducted in accordance with the Guidelines for the Care and Use of Laboratory Animals of Hunan University, and experiments were approved by the Animal Ethics Committee of the College of Biology (Hunan University).

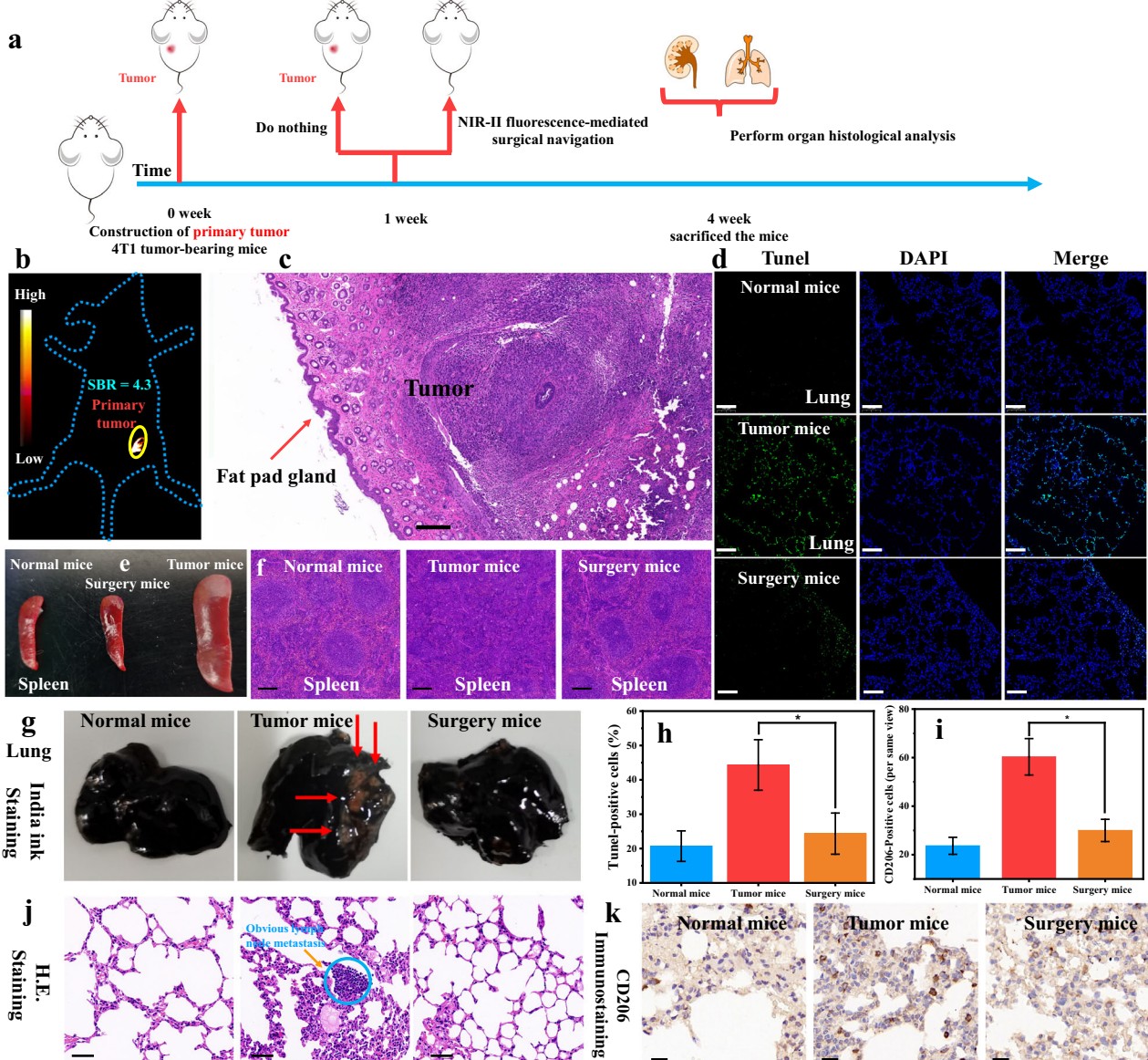

**Fig. 7 | Surgical navigation for a primary breast tumor and assessment of tumor metastasis. a** Schematic illustration showed the experiments of 4T1 tumor-bearing mice (the primary tumor). **b** NIR-II imaging in vivo before surgical navigation with **HPQ-Zzh-B**, $\lambda_{ex} = 808$ nm, collection channel: 1000–1700 nm. **c** Histology studies of tumor resected after NIR-II fluorescence-mediated surgery for primary tumor, scale bar: 200 µm. The experiment was repeated three times independently, with similar results. After surgical navigation, the primary tumor had only a small amount of fat pad gland. The histological studies indicated that NIR-II fluorescence, based on **HPQ-Zzh-B**, could effectively distinguish the boundary between normal and tumor tissues. **d** Tunel-positive apoptotic cells in the lungs of indicated mice, scale bar: 100 µm. **e** Photos of spleens of different mice, scale bar: 3 mm. Mice with primary tumors had larger spleens, while the spleens of mice resected by NIR-II fluorescence-mediated surgery were similar in size to normal mice. **f** Histology studies of indicated mice for spleens, scale bar: 200 µm. The experiment was repeated three times independently, with similar results. The spleens of mice with primary tumors had a vigorous immune response, and the spleens of mice resected

by NIR-II fluorescence-mediated surgery were similar to normal mice. **g** Images of lung tissue after being treated with India ink, the red arrow showed a pulmonary tumorous node, scale bar: 2 mm. This is an evaluation of lung metastasis inhibition efficacy on the NIR-II fluorescence-mediated surgical navigation. **h** for **d**. Data were presented as mean ± s.d. derived from $n = 3$ independent biological samples, *$p = 0.026$, *$p < 0.05$, and statistical significance was assessed via an unpaired two-sided student $t$-test. **i** for **k**. Data were presented as mean ± s.d. derived from $n = 3$ independent biological samples, *$p = 0.042$, *$p < 0.05$, and statistical significance was assessed via an unpaired two-sided student $t$-test. **j** Histology studies of indicated mice for lungs, the blue circle showed a pulmonary tumorous node, scale bar: 50 µm. The experiment was repeated three times independently, with similar results. The lungs of mice with primary tumors showed obvious tumor metastasis, and the lungs of mice resected by NIR-II fluorescence-mediated surgery were similar to normal mice. **k** The CD206 immunohistochemistry in the lungs of the indicated animals, scale bar: 20 µm. The CD206 is a protein associated with tumor metastasis, and its increased content means tumor metastasis.

## Octanol/water partition-coefficient (log P$_{o/w}$)

We used the "shake-flask" method to determine the octanol/water partition-coefficient. We mixed the water with octanol and shook it thoroughly to equilibrium, which made the two phases saturated with each other. The two layers were separated for subsequent experiments. Subsequently, we dissolved the compounds into methanol and drew the standard absorption curve. A certain

volume of octanol saturated with water was taken to dissolve the compound, and an equal volume of water saturated with octanol was added. The above mixture was shaken at room temperature for 24 h, the two phases were then carefully separated. Using the extinction coefficient of the compounds in octanol saturated with water, the concentration of the compounds was determined by UV-visible spectroscopy. The octanol/water partition-coefficient

was calculated by the following formula:

$$\log P_{O/W} = \log\left(\frac{c_{octanol}}{c_{water}}\right)$$

### 4T1 cells culture
4T1 cells were cultured in high glucose Dulbecco's Modified Eagle Medium (DMEM, Hyclone) supplemented with 10% fetal bovine serum (FBS, BI), and 1% antibiotics (100 U/mL penicillin and 100 μg/mL streptomycin, Hyclone) at 37 °C and 5% $CO_2$. Cells were carefully harvested and split when they reached 80% density to maintain exponential growth.

### ICG-J-aggregates
Use deionized water and ICG solid powder to make a 1.5 mM ICG aqueous solution. 1.5 mM ICG aqueous solution was heated at 66 °C for 24 h. The process is monitored with a UV-vis-NIR spectrophotometer. After the formation of J-aggregates, the solution was dialyzed against DI water for 24 h to remove any possible free ICG.

**AFM imaging.** About 10 μL sample (500 nM, dissolved in EtOH) was dropped on a freshly cleaved mica surface. After drying thoroughly (No liquid at all), the samples were scanned with Bruker Dimension Icon Scanning Probe Microscope (Bruker, Germany).

**SAXS test**. Take some samples and place them on the sample table for the sample test. Copper target: optical tube power 30 W, wavelength 1.54189 Å, the detector Pilatus 3 R 300 K, single pixel size 172 μm.

### Agarose gel anti-diffusion experiments

1. Prepare 3% (mass fraction) agarose aqueous solution (100 mL).
2. The configured aqueous agarose solution was heated in a microwave oven for 1 min.
3. Add the heated aqueous agarose solution to two quartz Petri dishes (50 mL, respectively).
4. The quartz Petri dishes were then placed in the refrigerator (4 °C) for 10 min, and we could gain two pieces of agarose gel.
5. A circular mold was used to draw the circle on the agarose gel, and the fluorophore was subsequently added to the circle.
6. After NIR-II imaging at the initial conditions, a small amount of water was added to the agarose, followed by NIR-II imaging over time.

### Culture conditions for the mice
Mice (8-week-old) were purchased from Hunan Slake Jingda Laboratory Animal Co., Ltd (China). Mice were housed under controlled conditions (22 °C, 55 - 65% humidity, 12 h light-dark cycle) and were allowed free access to tap water.

### 4T1 tumor-bearing mice, subcutaneous tumor
All animal experiments were performed in compliance with the relevant laws and approved by the Institutional Animal Care and Use Committee of Hunan University. Female balb/c nude mice were implanted subcutaneously with 50 μL of PBS containing 4T1 cancer cells ($2 \times 10^6$) to develop a tumor model.

### 4T1 tumor-bearing mice, primary tumor
All animal experiments were performed in compliance with the relevant laws and approved by the Institutional Animal Care and Use Committee of Hunan University. About 50 μL of PBS containing 4T1 cancer cells are surgically implanted into the mammary fat pads of anesthetized female balb/c mice by an L-shaped incision (1 cm × 1 cm) between the abdominal midline and the fourth and fifth nipples. Once the skin flap is opened with moist cotton swabs and the mammary fat pads are exposed, cells are injected into the fat pads with an insulin syringe at a maximum of 50 μL of volume.

### Surgical navigation
According to the clinical strategy of surgical resection of breast cancer, we injected the probe (HPQ-Zzh-B: 100 μM, 25 μL, pH 7.4 DPBS containing 10% EtOH and 1% Tween), specifically into the tumor to delimit the boundary (intra-tumoral injection). After waiting for NIR-II fluorescence signal stabilization, tumor resection surgery guided by NIR-II fluorescence was performed.

### Histology studies
The organs, muscles, and tumors of female balb/c mice were fixed in 10% formaldehyde immediately after sacrifice. Histological examination was according to a conventional method and stained with hematoxylin and eosin (H&E). The nucleus is blue and the cytoplasm is red. Classify and record the morphology of any observed lesions according to classification criteria.

### Staining mice's lungs with India ink
All animal experiments were performed in compliance with the relevant laws and approved by the Institutional Animal Care and Use Committee of Hunan University. The lungs of female balb/c mice were injected with Indian ink along the trachea after sacrifice.

### Immunohistochemical experiments of paraffin section

1. Deparaffinzing and rehydrating the paraffin section: put the sections into a dewaxing solution for 15 min (I)—dewaxing solution for 15 min (II)—dewaxing solution for 15 min (III)—ethanol for 5 min (IV)—absolute ethanol for 5 min (V)—ethanol for 5 min (VI)—rinse in distilled water (VII).
2. Antigen retrieval: The tissue sections are placed in a repair box filled with citric acid (pH 6.0) antigen retrieval buffer for antigen retrieval in a microwave oven, heated on medium power for 8 min until boiling, then turned off the microwave oven, kept warm for 8 min and then transferred to medium-low power for heating 7 min. During this process, excessive evaporation of buffer should be prevented, and the sections should not be allowed to dry. To cool to room temperature before proceeding, the sections are placed in PBS (pH 7.4) and shaken on the decolorization shaker three times for 5 min each.
3. Blocking endogenous peroxidase activity: the sections are placed in 3% hydrogen peroxide and incubated at room temperature in darkness for 25 min. The sections are placed in PBS (pH 7.4) and shaken on a decolorizing shaper three times for 5 min each.
4. Serum sealing: 3% BSA was added to the circle to evenly cover the tissue, and the tissues are sealed for 30 min at room temperature. (Primary antibody is sealed with normal rabbit serum from goat source and other sources are sealed with BSA).
5. Primary antibody (Anti-Mannose Receptor/CD206 Rabbit pAb, GB113497, dilute with the Primary Antibody Dilution Buffer, G2025) incubation: the sealing solution is gently removed, the primary antibody prepared with PBS (pH 7.4) in a certain proportion (1: 400) is added to the sections, and the sections are placed flat in a wet box and incubated overnight at 4 °C. (Add a small amount of water in the wet box to prevent evaporation of antibodies).
6. Secondary antibody (HRP conjugated Goat Anti-Rabbit IgG (H + L), GB23303) incubation: the sections are placed in PBS (pH 7.4) and washed by shaking on the decolorizing shaker three times for 5 min each. After the sections are slightly shaken and dried, the tissues are covered with secondary antibody (HRP labeled, 1:500 in PBS) from the corresponding species of primary antibody and incubated at room temperature for 50 min.

7. DAB chromogenic reaction: the sections are placed in PBS (pH 7.4) and shaken on the decoloring shaker three times for 5 min each. DAB color-developing solution newly prepared is added in the circle after the sections are slightly dried. The color developing time is controlled under the microscope. The positive is brownish yellow. Rinse the sections with tap water to stop the reaction.

8. Nucleus counterstaining: the sections are counterstained with hematoxylin stain solution for about 3 min; washed with tap water; differentiated with hematoxylin differentiation solution for several seconds; washed with tap water; treated with hematoxylin returning blue solution; washed with running water.

9. Dehydration and mounting: place the section in 75% alcohol for 5 min (I)−85% alcohol for 5 min (II)−absolute ethanol for 5 min (III) −absolute ethanol for 5 min (IV)−n-butanol for 5 min (V)− xylene 5 min (VI), dehydrated and transparent, remove the sections from xylene and let them dry slightly, then mount the sections with neutral gum.

10. Visualize staining of tissue under a microscope, acquisitive and analysis image.

11. The nucleus of the hematoxylin stained is blue, and the positive expression of DAB is brownish yellow.

## Immunofluorescence experiments of paraffin section

1. Deparaffinizing and rehydrating the paraffin section: put the sections into a dewaxing solution for 15 min (I)−dewaxing solution for 15 min (II)−dewaxing solution for 15 min (III)−ethanol for 5 min (IV)−absolute ethanol for 5 min (V)−ethanol for 5 min (VI)− rinse in distilled water (VII).

2. Antigen retrieval: immerse the slides in EDTA antigen retrieval buffer (pH 8.0) and maintain them at a sub-boiling temperature for 8 min, standing for 8 min, and then followed by another sub-boiling temperature for 7 min. Be sure to prevent the buffer solution from evaporating. Let the air cool. Wash three times with PBS (pH 7.4) in a Rocker device, 5 min each. Use the right antigen retrieval buffer and heat extent according to tissue characteristics.

3. Circle and serum blocking: eliminate obvious liquid and mark the objective tissue with a liquid blocker pen. Add 3% BSA to cover the marked tissue to block non-specific binding for 30 min. Cover the objective area with 10% donkey serum (for the case of primary antibody originated from goat) or 3% BSA (for the case of primary antibody originated from others).

4. Primary antibody: throw away the blocking solution slightly. Incubate slides with primary antibody (diluted with PBS appropriately) overnight at 4 °C, placed in a wet box containing a little water.

5. Secondary antibody: wash slides three times with PBS (pH 7.4) in a Rocker device, 5 min each. Then throw away the liquid slightly. Cover objective tissue with secondary antibody (appropriately respond to primary antibody in species) and incubate at room temperature for 50 min in dark conditions.

6. DAPI counterstain in nucleus: wash three times with PBS (pH 7.4) in a Rocker device, 5 min each. Then incubate with DAPI solution at room temperature for 10 min, kept in the dark place.

7. Spontaneous fluorescence quenching: wash three times with PBS (pH 7.4) in a Rocker device, 5 min each. Add spontaneous fluorescence quenching reagent to incubate for 5 min. Wash in running tap water for 10 min.

8. Mount: Throw away liquid slightly, then cover slip with an anti-fade mounting medium.

9. Microscopy detection and collect images by Fluorescent Microscopy. DAPI glows blue by UV excitation wavelength 330−380 nm and emission wavelength 420 nm; FITC glows green by excitation wavelength 465−495 nm and emission wavelength 515−555 nm.

10. The nucleus is blue by labeling with DAPI. Positive cells are green according to the fluorescent labels used.

### Reporting summary

Further information on research design is available in the Nature Portfolio Reporting Summary linked to this article.

## Data availability

The authors declare that other data related to this research are available within the paper and its Supplementary Information or from the authors upon request. Source data are provided with this paper.

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

## Acknowledgements

This work is supported by the National Natural Science Foundation of China (22234003, X.-B.Z. and 22004033, T.-B.R.) and Special Funds for the Construction of Innovative Provinces in Hunan Province (2019RS1031, X.-B.Z.).

## Author contributions

Z.L., T.-B.R., L.Y., and X.-B.Z. conceived and designed the study. Z.L. and P.-Z.L. completed organic synthesis. Z.L., P.-Z.L., L.X., X.-X.Z. K.L., Q.W., and X.-F.L. conducted animal experiments. Z.L. conducted data analysis. Z.L. and T.-B.R. wrote the manuscript. L.Y. and X.-B.Z critically revised the manuscript. All of the authors approved the final version of the manuscript.

## Competing interests

The authors declare no competing interests.
