## [Peer Review File · Nature Communications]

In Situ Orderly Self-assembly Strategy Affording NIR-II-J-Aggregates for In Vivo Imaging and Surgical NavigationReviewers' Comments:

Reviewer #1:

Remarks to the Author:

In this manuscript, the authors reported the successful synthesis of a series of HPQ based J-aggregates with bright NIR-II fluorescence. As a proof-of-concept study, the authors found that the ONOO⁻ activatable HPQ derivatives of HPQ-Zzh-B upon tumor accumulation could specifically light up tumors in response to intratumoral ONOO⁻. Therefore, such HPQ-Zzh-B based NIR-II-J-aggregate probe was confirmed to be able to distinguish tumor tissues from adjacent normal tissues, thereby enabling precise primary tumor resection by NIR-II imaging navigation. Overall, this manuscript is well-prepared, and is a good contribution to the field of imaging guided tumor surgery.

Comments:

1. As shown in figure 5, to confirm the ONOO⁻ responsiveness of HPQ-Zzh-B, HPQ-LZ and HPQ-Zzh are suggested to be included in the experiments as the negative and positive controls, respectively. In addition, the color bar for in vivo fluorescence images should be provided.
2. The authors demonstrated that the HPQ-Zzh probe exhibited excellent antidiffusion behaviors and thus would stay at the injected tumors for a long time as shown in Figure 5i. Considering such probe was specifically injected into tumors while not systemic administration, I think that such probe could not evenly and selectively distribute inside whole tumor tissue based on our previous experimental experience. Actually, in my opinion, the activatable probe with reasonable diffusion capacity would be more suitable to intratumoral injection and subsequent imaging guided surgical resection because the probe can only be activated by the intratumoral microenvironments.
3. Font size of these inset figures and legends was too small to read, please reformat these figures.
4. The authors are suggested to polish the English writing of the manuscript.

Reviewer #2:

Remarks to the Author:

The manuscript by Ke Li et al. reports on the design of solid-state fluorophores based on 2-(2-Hydroxyphenyl)-4(3H)-quinazolinone (HPQ) derivatives to obtain stable NIR-II-J-aggregates in vivo with low or anti diffusion properties which allows selective imaging of the tumor as well as its precise resection.

The most successful compound (HPQ-Zzh-B) consisted in fusing a HPQ unit onto the conjugated structure of a simple hemi-cyanine dye and protecting the hydroxyl group with phenylboronic acid pinacol ester which is ONOO⁻ sensitive group. HPQ-Zzh-B was not fluorescent in physiological conditions, however upon removing of the protecting group by ONOO⁻ (high concentration in tumor), the released dye molecules were able to form J-aggregates with strong NIR-II fluorescence.

This is a well executed study and although similar strategy has been already reported by the authors but with another activatable group (Li et al. PNAS 2021, <https://doi.org/10.1073/pnas.2018033118>), I think the work is of great interest. However, I have some concerns that should be carefully addressed by the authors before any decision. Therefore I recommend major corrections. My comments are listed below:

1. Since the authors have published already similar compounds that can form J-aggregates with enhanced activatable fluorescence in situ in a previous work, they should explain more what is the additional value of the synthesized compounds compared to previous one.
2. The materials and methods section is missing in the manuscript. I found only a short paragraph about the Materials and general methods. Cell culture in the main manuscript?!!! Indeed, I did not find neither in the manuscript nor in the SI, the illumination conditions in vivo. What is the fluence of the

light source? What kind of laser? What was the wavelength of excitation and that of emission? In which solvent the fluorescent probe was injected in vivo? How it is administered SC, IV...?

3. Authors tried to compare the performance of their fluorophores in terms of J-aggregation, stability and anti diffusion properties to other dyes previously described. However, authors should describe/define what are the properties/characteristic of these dyes. Also authors should explain why they choose these dyes for comparison.

4. The authors reported that no concentration dependence on the optical properties of HPQ-zzh, if so, then how these J-aggregates are formed? Only dimers were formed? In fact, I was wondering if these J-aggregates are based on supramolecular assemblies of the monomers in water.

5. For the stability study, authors used first Dulbecco's phosphate buffered saline (DPBS) and 1% Tween to simulate the physiological environment. Why the authors used such condition? Afterwards, authors claimed that they added Fetal Bovine Serum FBS (10 % vol) did not affect the stability of the aggregates. Does the addition of FBS was done on the same suspension of DPBS and 1% of tween or on new suspension. Also I do not understand why the authors compared the stability of the J-aggregates of different components but in different conditions as shown in figure 4! It ma be useful that authors precise why they investigated the stability of HP-Zzh, HPQ-Zzh and ICG in DPBS +1% Tween but only water for ICG-J-aggregates?

6. Did the authors assess the structure of the formed aggregates of HPQ-Zzh or HPQ-Zzh-B upon their reaction with ONOO- using AFM, Cryo-TEM or SAXS for example? This will give better understanding on how these molecules are organized inside the aggregates.

7. I did not understand what the authors want to say by the following sentence: "Therefore, we excluded the possibility of J-aggregation owing to large concentrations for HPQ-Zzh, which also confirmed the stability of the J-aggregated form of HPQ-Zzh." This should be clarified in the main text.

8. Authors should be more precise in their description of other dyes " DAD-7401, 6 (good lipophilicity) and ICG (good hydrophilicity) were selected as the reference fluorophores.". What does it mean good lipophilicity? Is their any values range to consider if this is good or bad lipophilicity and hydrophilicity.

9. I was wondering about the utility of investigating the anti diffusion properties on TLC Fig. S21/ S22), since the diffusion of compounds depends mainly on the polarity of the chemical compound, its solubility in the chosen solvent but also on its concentration. Please precise how the experiment on agarose gel was performed?

10. The quality of some figures should be improved because some details can be barely seen. Figure 3e and f: please used full color for the histograms. Figures 4 e and f: Please enlarge the font size above the red shifted spectra, also enlarge the inset or move it to the SI.

Minor points:

Figure 3 : "partikal" should be "particle"

To rewrite the following sentence: " Firstly, we studied self-assembly form stability of NIR-II-J-aggregates"

Please improve the quality of figure s9, mainly enlarge the title of each graph.

Please define: HD-B probe

Please use the term photobleached instead of destroyed in the following sentence: "Under 808 nm laser irradiation, the absorption peak shape of HPQ-Zzh remained unchanged, while ICG and the ICG-J-aggregates were destroyed"

Response to Reviewer #1:

In this manuscript, the authors reported the successful synthesis of a series of HPQ based J-aggregates with bright NIR-II fluorescence. As a proof-of-concept study, the authors found that the ONOO⁻ activatable HPQ derivatives of HPQ-Zzh-B upon tumor accumulation could specifically light up tumors in response to intratumoral ONOO⁻. Therefore, such HPQ-Zzh-B based NIR-II-J-aggregates probe was confirmed to be able to distinguish tumor tissues for adjacent normal tissues, thereby enabling precise primary tumor resection by NIR-II imaging navigation. Overall, this manuscript is well-prepared, and is a good contribution to the field of imaging guided tumor surgery.

Response: We sincerely appreciate the positive comments from the reviewer. According to the comments, we have revised the manuscript carefully. Our point-by-point answers are as follows:

1. As shown in figure 5, to confirm the ONOO⁻ responsiveness of HPQ-Zzh-B, HPQ-LZ and HPQ-Zzh are suggested to be included in the experiments as the negative and positive controls, respectively. In addition, the color bar for in vivo fluorescence images should be provided.

Response: We thank the reviewer for the suggestion. We have added the relevant control experiments (the ONOO⁻ responsiveness of HPQ-LZ and HPQ-Zzh) into the revised manuscript, as shown in Figure S24 in the revised supporting information and we have made corresponding revisions in the page 5 of revised manuscript. The results showed that, unlike HPQ-Zzh-B, ONOO⁻ did not cause the absorption redshift of HPQ-LZ and HPQ-Zzh. Meanwhile, the color bars for in vivo fluorescence images were added in the revised manuscript, as shown in Figure 5, 6, 7.

Figure S24 | Normalized absorbance spectra of (a) HPQ-LZ (10 μM) and (b) HPQ-Zzh (10 μM) in the absence and presence of (5 μM) ONOO⁻ in DPBS +1% Tween.

2. The authors demonstrated that the HPQ-Zzh probe exhibited excellent anti-diffusion behaviors and thus would stay at the injected tumors for a long time as shown in Figure 5i. Considering such probe was specifically injected into tumors while not systemic administration, I think that such probe could not evenly and selectively distribute inside whole tumor tissue based on our previous experimental experience. Actually, in my

opinion, the activatable probe with reasonable diffusion capacity would be more suitable to intra-tumoral injection and subsequent imaging guided surgical resection because the probe can only be activated by the intra-tumoral microenvironments.

Response: We are sorry that it was not clearly stated in the last manuscript.

Generally, the HPQs-based probes are constructed by introducing a substrate of targets into the phenolic hydroxy groups of HPQ-analogs, which prohibit the H-bonding interactions. Thus, like conventional small molecular fluorescent probes, the activatable probe HPQ-Zzh-B is soluble under physiological condition, and can distribute at the tumor site by diffusion. However, after the hydroxyl protective group on the probe HPQ-Zzh-B was removed by the analytes, the released dye HPQ-Zzh were immediately self-assembled in situ to form anti-diffusion nanoaggregates with NIR-II emission, enabling them to remain in the cells and achieve long-term tracking.

3. Font size of these inset figures and legends was too small to read, please reformat these figures.

Response: We thank the reviewer for the suggestion. We have enlarged the font size in inset figures and legends in the revised manuscript. Please see Figure 3 and Figure 4 in the revised main text.

4. The authors are suggested to polish the English writing of the manuscript.

Response: Thank you for your valuable and thoughtful comments. We have carefully checked and improved the English writing in the revised manuscript.

Response to Reviewer #2:

The manuscript by Zhe Li et al. reports on the design of solid-state fluorophores based on 2-(2-Hydroxyphenyl)-4(3H)-quinazolinone (HPQ) derivatives to obtain stable NIR-II-J-aggregates in vivo with low or anti-diffusion properties which allows selective imaging of the tumor as well as its precise resection. The most successful compound (HPQ-Zzh-B) consisted in fusing a HPQ unit onto the conjugated structure of a simple hemi-cyanine dye and protecting the hydroxyl group with phenylboronic acid pinacol ester which is ONOO⁻ sensitive group. HPQ-Zzh-B was not fluorescent in physiological conditions, however upon removing of the protecting group by ONOO⁻ (high concentration in tumor), the released dye molecules were able to form J-aggregates with strong NIR-II fluorescence. This is a well-executed study and although similar strategy has been already reported by the authors but with another activatable group (Li et al. PNAS 2021, <https://doi.org/10.1073/pnas.2018033118>), I think the work is of great interest. However, I have some concerns that should be carefully addressed by the authors before any decision. Therefore, I recommend major corrections.

Response: We sincerely appreciate the comments from the reviewer. Our point-by-point answers to the suggestions are listed below:

1. Since the authors have published already similar compounds that can form J-aggregates with enhanced activatable fluorescence in situ in a previous work, they

should explain more what is the additional value of the synthesized compounds compared to previous one.

Response: We sincerely appreciate the reviewer's suggestion. In 2021, our group reported a HPQ analogue (HYPQ), and used it to design anti-diffusion probe HYPQG for in situ imaging of the membrane-associated enzyme, glutamyl transpeptidase (GGT). However, the absorption and emission wavelengths of HYPQ are less than 600 nm and 700 nm respectively, which limits the penetration of biological tissues and the signal-to background ratios (SBRs) for in vivo imaging. In contrast, short wave infrared (SWIR or NIR-II, 900–1700 nm) imaging has attracted widespread interest in the past decade owing to its deeper penetration of biological tissues, reduced autofluorescence, and improved SBRs. In this study, we introduced positively charged electron withdrawing group to transform existing HPQ into proper J-aggregates-based anti-diffusion NIR-II HPQs scaffolds (HPQ-LZ and its derivatives, $\lambda_{ex}/\lambda_{em} = 808\text{ nm}/960\text{ nm}$) for NIR-II imaging. Through the regulation of hydroxyl group, an anti-diffusion activatable NIR-II ONOO⁻ probe HPQ-Zzh-B was developed. Using the probe HPQ-Zzh-B, we not only long-term visualized the tumour, but also realized in situ NIR-II imaging-guided surgery. We have made corresponding revisions in the DISCUSSION section of revised manuscript. Please see page 7.

2. The materials and methods section is missing in the manuscript. I found only a short paragraph about the Materials and general methods. Cell culture in the main manuscript!!!! Indeed, I did not find neither in the manuscript nor in the SI, the illumination conditions in vivo. What is the fluence of the light source? What kind of laser? What was the wavelength of excitation and that of emission? In which solvent the fluorescent probe was injected in vivo? How it is administered SC, IV...?

Response: We apologize for putting such information (materials and methods section) in an unobtrusive position (Support Information) in last manuscript. Relevant information has been added to the revised manuscript, please see yellow background in Page 8-10 in the revised manuscript, as shown in **Methods** section.

3. Authors tried to compare the performance of their fluorophores in terms of J-aggregation, stability and anti-diffusion properties to other dyes previously described. However, authors should describe/define what are the properties/characteristic of these dyes. Also, authors should explain why they choose these dyes for comparison.

Response: We thank the reviewer for the suggestion. According to the suggestion of review, we have described these dyes as follow:

Indocyanine green (ICG) is a commercial and widely-used cyanine dye for FDA-approved application in humans and its J-aggregates have been well studied now (ref: Nanotheranostics 2017, 1, 430-439, Nat. Commun. 2021, 12, 5410, etc.). FD-1080 is a cyanine dye with NIR-II fluorescence and its J-aggregates is one of the rare NIR-II J-aggregates in recent years (ref: J. Am. Chem. Soc. 2019, 141, 19221-19225). DAD-740 is a classic type of organic NIR-II fluorophores used for in vivo imaging (ref: Nat. Mater. 2016, 15, 235-242).

As shown in Figure 4g, FD-1080 J-aggregates show carrier and concentration

dependence, which are same to traditional J-aggregates. Although ICG J-aggregates are not carrier-dependent *in vitro*, they are easily disassembled in the presence of hydrophobic biomolecules (as shown in Figure 4i-j), which is also other type's J-aggregates features. Therefore, ICG J-aggregates and FD1080 J-aggregates were chosen for comparison, which could better illustrate the self-assembly form stability of novel NIR-II-J-aggregates (HPQ-LZs) compared to conventional J-aggregates.

On the other hand, as a classic cyanine dye, ICG ($\log P_{o/w} = 0.502$) is water-soluble due to its two sulfonic acid groups and one positive charge. As a benzobisthiadiazole-type NIR-II dye, DAD-740 ($\log P_{o/w} = 3.308$) has good lipophilicity in organic solvents due to its alkyl chain. Hence, we selected ICG and DAD-740 to conduct the anti-diffusion control experiments, which could fully reflect the novel NIR-II-J-aggregates (HPQ-LZ and HPQ-Zzh) good anti-diffusion ability.

4. The authors reported that no concentration dependence on the optical properties of HPQ-Zzh, if so, then how these J-aggregates are formed? Only dimers were formed? In fact, I was wondering if these J-aggregates are based on supramolecular assemblies of the monomers in water.

Response: We sincerely appreciate the reviewer for pointing out this issue. I'm sorry that, in last manuscript, it is not adequately accurate that HPQ-Zzh have no concentration dependence. We revised the description in the manuscript, that is, the novel NIR-II-J-aggregates have lower-concentration dependence.

According to original HPQ's single crystal structure (as shown in Figure 2a), we know that HPQ is a solid-state fluorescent dye based on H-bonding self-assembly and the hydrogen atom on the amide is the key to dimer formation. In order to figure out whether the novel NIR-II HPQs have similar structure, we synthesized the reference compound HPQ-LZ-Me (methyl substituted for amide hydrogen) and tested its spectrum in THF. The experimental results showed that, compared with HPQ-LZ ($\lambda_{ab}/\lambda_{em} = 750/960$ nm), HPQ-LZ-Me ($\lambda_{ab}/\lambda_{em} = 470/620$ nm, similar to monomers) did not form NIR-II-J-aggregates (as shown in Figure S6), which indicated eliminating the hydrogen atom on the amide, could effectively inhibit NIR-II-J-aggregates formation. On the basis, we speculated that such NIR-II-J-aggregates had the similar self-assembly mechanism to the original HPQs, thus they were based on supramolecular assemblies of the dimers.

We have made corresponding revisions in page 3 of revised manuscript.

Figure S6 | Normalized absorbance and fluorescence spectrum of HPQ-LZ-Me (20 μ M) and HPQ-LZ (20 μ M) in THF, $\lambda_{ex} = 450$ nm for HPQ-LZ-Me, $\lambda_{ex} = 808$ nm for HPQ-

LZ.

5. For the stability study, authors used first Dulbecco's phosphate buffered saline (DPBS) and 1% Tween to simulate the physiological environment. Why the authors used such condition? Afterwards, authors claimed that they added Fetal Bovine Serum FBS (10 % vol) did not affect the stability of the aggregates. Does the addition of FBS was done on the same suspension of DPBS and 1% of tween or on new suspension. Also, I do not understand why the authors compared the stability of the J-aggregates of different components but in different conditions as shown in figure 4l? It may be useful that authors precise why they investigated the stability of HP-Zzh, HPQ-Zzh and ICG in DPBS +1% Tween but only water for ICG-J-aggregates?

Response: We appreciate the reviewer's careful thoughts over this point.

As is well known, cell contain two immiscible phases, oil phase and water phase (including inorganic salts). DPBS is a buffered solution containing inorganic salts, which is often utilized in cell culture. While tween, a non-ionic surfactant, can form micelles in aqueous solution, which can simulate the lipophilic structure in cells. Therefore, DPBS + 1% Tween were used to simulate the physiological environment. (We have made corresponding revisions in page 4 of revised manuscript.) In addition, we also tested the absorption spectra of HPQ-Zzh at different concentrations in DPBS + 10% FBS (and DPBS + 1% Tween + 10% FBS). Experimental results showed that, although the concentrations of HPQ-Zzh varied, the absorption peak shape of HPQ-Zzh was not found to change, indicating that HPQ-Zzh only had lower-concentration dependence (as shown in Figure 4f and S14).

In the manuscript, we tested the stability of related compounds (HP-Zzh, HPQ-Zzh, ICG and ICG J-aggregates). We chose DPBS + 1% Tween as the solvent to test the stability of HP-Zzh, HPQ-Zzh and ICG, because DPBS + 1% Tween could simulate physiological environment. Meanwhile, we found that ICG J-aggregates depolymerized in the DPBS + 1% Tween (as shown in Figure 4i-j), due to its poor self-assembly stability. Therefore, to exclude the effects of poor self-assembly stability as much as possible, we selected a solvent (water) in which ICG J-aggregates self-assembly form is stable, to carried out the photostability and chemical stability experiments.

6. Did the authors assess the structure of the formed aggregates of HPQ-Zzh or HPQ-Zzh-B upon their reaction with ONOO⁻ using AFM, Cryo-TEM or SAXS for example? This will give better understanding on how these molecules are organized inside the aggregates.

Response: We thank the reviewer for the suggestion. We performed SAXS and AFM experiments of HPQ-Zzh (as shown in Figure S22). Due to the limitation of experimental conditions, Cryo-TEM experiment was not performed. We firstly explored the internal arrangement of HPQ-Zzh through SAXS experiment. Experimental results showed that the SAXS atlas of HPQ-Zzh had some sharp peaks (not broad peaks), which may be due to highly ordered molecular arrangement of NIR-II-J-aggregates based on H-bonding interactions (In the SAXS atlas, the more pointed peaks represent higher crystallinity and more regular molecular arrangement). Further,

we characterized the morphology of the NIR-II-J-aggregates by using atomic force microscopy (AFM). The drop-casting of the EtOH solution of HPQ-Zzh J-aggregates onto the surface of a silica wafer resulted in the deposition of round cake-like aggregates (Fig. S22) with a length of 550 ± 20 nm, a width of 550 ± 20 nm, and a height of 85 ± 5 nm, respectively. The relatively regular surface morphology may be related to the orderly stacking structure of J-aggregates. In addition, previous studies have indicated the novel NIR-II-J-aggregates displayed a similar self-assembly mechanism to the original HPQs, thus we concluded our novel NIR-II-J-aggregates were based on the supramolecular assemblies of the dimers.

We have made corresponding revisions in page 5 of revised manuscript.

Figure S22 | (a) One-dimensional SAXS atlas, (b) Two-dimensional SAXS pattern of HPQ-Zzh, and (c) AFM results of HPQ-Zzh, scale bar: 1 μm .

7. I did not understand what the authors want to say by the following sentence: “Therefore, we excluded the possibility of J-aggregation owing to large concentrations for HPQ-Zzh, which also confirmed the stability of the J-aggregated form of HPQ-Zzh.” This should be clarified in the main text.

Response: We sincerely appreciate the reviewer for pointing out this issue.

We have made the corresponding revisions. “Therefore, we excluded the possibility of J-aggregation owing to high concentrations for HPQ-Zzh, which also confirmed the stability of the J-aggregated form of HPQ-Zzh.” have been changed to “The experimental results showed that the absorption peaks of HPQ-Zzh at different concentrations (1-50 μM) were basically consistent, indicating that the J-aggregates of HPQ-Zzh have good self-assembly stability and lower concentration dependence (even 1 μM).” We have made corresponding revisions in the page 4 of revised manuscript.

8. Authors should be more precise in their description of other dyes “DAD-740 (good lipophilicity) and ICG (good hydrophilicity) were selected as the reference fluorophores.”. What does it mean good lipophilicity? Is their any values range to consider if this is good or bad lipophilicity and hydrophilicity.

Response: We sincerely appreciate the reviewer for pointing out this issue. We have given relevant descriptions of DAD-740 and ICG in Question 3, and made corresponding modifications in the revised manuscript.

Lipophilicity refers to the ability of compounds to dissolve in fats, oils, lipids or nonpolar solvents, and hydrophilicity refers to the ability of compounds to dissolve in

a water environment (ref: Front. Cardiovasc. Med. 2021, 8, 687585). Of note, lipophilicity (and hydrophilicity) is commonly described by $\log P_{o/w}$ (oil/water partition coefficient), which is calculated using the shake flask method (ref: Angew. Chem. Int. Ed. 2016, 55, 3858-3860). It is easy to understand that, the larger the $\log P_{o/w}$, the better lipophilicity for a compound, otherwise, it has the more hydrophilic properties. In general, $\log P_{o/w} > 3$ is considered as good-lipophilicity compounds (The compound is 1000 times more soluble in n-octanol than water). And $\log P_{o/w} < 1$ is considered to have a certain degree hydrophilicity. $\log P_{o/w}$ calculations showed that the values of DAD-740 was 3.308 and ICG was 0.502, indicating DAD-740 had good lipophilicity and ICG had good hydrophilicity. We have made corresponding revisions in the page 5 of revised manuscript.

9. I was wondering about the utility of investigating the anti-diffusion properties on TLC Fig. S21/ S22), since the diffusion of compounds depends mainly on the polarity of the chemical compound, its solubility in the chosen solvent but also on its concentration. Please precise how the experiment on agarose gel was performed?

Response: We agree with the reviewer that the anti-diffusion properties of compounds on TLC are generally related to their polarity, solubility and concentration. Considering above, the anti-diffusion experiments on TLC are inappropriate, which does not fully demonstrate the anti-diffusion properties of the compounds. Therefore, we have removed the description of the corresponding part in the manuscript. The detailed steps of agarose gel anti-diffusion experiments are as shown in method sections (Page 8).

10. The quality of some figures should be improved because some details can be barely seen. Figure 3e and f: please used full color for the histograms. Figure 4 e and f: Please enlarge the font size above the red shifted spectra, also enlarge the inset or move it to the SI.

Response: We thank the reviewer for the suggestion. In the manuscript, we have made corresponding modifications, as shown in Figure 3e-f, Figure 4 e-f and Figure S14.

Minor points:

11. Figure 3: “partikal” should be “particle”; To rewrite the following sentence: “Firstly, we studied self-assembly form stability of NIR-II-J-aggregates”

Response: We thank the reviewer for the suggestion. In the manuscript, we have made corresponding modifications. In Figure 3: “partikal” have been changed to “particle”. “Firstly, we studied self-assembly form stability of NIR-II-J-aggregates” have been converted to “To demonstrate that the novel NIR-II-J-aggregates could be utilized for bioimaging, we firstly investigated self-assembly stability of HPQ-Zzh.” As shown in page 4 of revised manuscript.

12. Please improve the quality of figure s9, mainly enlarge the title of each graph. Please define: HD-B probe. Please use the term photobleached instead of destroyed in the following sentence: “Under 808 nm laser irradiation, the absorption peak shape of HPQ-Zzh remained unchanged, while ICG and the ICG-J-aggregates were destroyed”

Response: We thank the reviewer for the suggestion. In the manuscript, we have made corresponding modifications. 1) Figure S9 (now Figure S10) has been made modifications. 2) The definition of HD-B has been revised at the corresponding location in the manuscript, as follows: HD-B probe (HD dye modified with Borate group) is a kind of widely used NIR-I ROS probe (J. Am. Chem. Soc. 2012, 134, 13510-13523, Figure S27). As shown in page 6 of revised manuscript. 3) “Under 808 nm laser irradiation, the absorption peak shape of HPQ-Zzh remained unchanged, while ICG and the ICG-J-aggregates were destroyed” have been converted to “Under 808 nm laser irradiation, the absorption peak shape of HPQ-Zzh remained unchanged, while ICG and the ICG-J-aggregates were photobleached” As shown in page 5 of revised manuscript.

Reviewers' Comments:

Reviewer #1:

Remarks to the Author:

I think this work with revisions is now acceptable for publication.

Reviewer #2:

Remarks to the Author:

I carefully read the revised version of the manuscript (NCOMMS-22-28309A) by Zhe Li and Coworkers entitled "In Situ Orderly Self-assembly Strategy Affording NIR-II-J-Aggregates for In Vivo Imaging and Surgical Navigation". The authors made a nice improvement of their data description and analysis compared to the first version. All of my previous questions were answered in a very convincing manner. Therefore I recommend the publication of the manuscript.